# Evaluating Seismic Beamforming Capabilities of Distributed Acoustic Sensing Arrays

Martijn P. A. van den Ende[1] and Jean-Paul Ampuero[1]

[1]Université Côte d'Azur, IRD, CNRS, Observatoire de la Côte d'Azur, Géoazur, France

**Correspondence:** M. van den Ende (martijn.vandenende@geoazur.unice.fr)

**Abstract.** The versatility and cost-efficiency of fibre-optic Distributed Acoustic Sensing (DAS) technologies facilitate geophysical monitoring in environments that were previously inaccessible for instrumentation. Moreover, the spatio-temporal data density permitted by DAS naturally appeals to seismic array processing techniques, such as beamforming for source location. However, the measurement principle of DAS is inherently different from that of conventional seismometers, providing measurements of ground strain rather than ground motion, and so the suitability of traditional seismological methods requires in-depth evaluation. In this study, we evaluate the performance of a DAS array in the task of seismic beamforming, in comparison with a co-located nodal seismometer array. We find that, even though the nodal array achieves excellent performance in localising a regional $M_L$ 4.3 earthquake, the DAS array exhibits poor waveform coherence and consequently produces inadequate beamforming results that are dominated by the signatures of shallow scattered waves. We demonstrate that this behaviour is likely inherent to the DAS measurement principle, and so new strategies need to be adopted to tailor array processing techniques to this emerging measurement technology. One strategy demonstrated here is to convert the DAS strain rates to particle velocities by spatial integration using the nodal seismmometer recordings as a reference, which dramatically improves waveform coherence and beamforming performance, and warrants new types of "hybrid" array design that combine dense DAS arrays with sparse seismmometer arrays.

## 1 Introduction

Dense seismmometer arrays play a central role in understanding various geological phenomena, including earthquake rupture behaviour (Kiser and Ishii, 2017; Meng et al., 2011), micro-seismicity (Inbal et al., 2016), fault zone structure (Zigone et al., 2019), and deep crustal and mantle geology (Jiang et al., 2018; Lin et al., 2013). Moreover, seismic arrays also serve civil protection purposes through monitoring nuclear test ban treaty violations (Ringdal and Husebye, 1982), monitoring volcano deformation and activity (Inza et al., 2011; Nakamichi et al., 2013), and potentially issuing earthquake early warnings (Meng et al., 2014). While the benefits of seismic arrays are evident, the deployment and maintenance of these arrays is (logistically) costly, and consequently they are often deployed as part of temporary campaigns.

The recent emergence of fibre-optic Distributed Acoustic Sensing (DAS; Hartog, 2017; Zhan, 2020) has opened up a plethora of possibilities and applications in seismic- and transient deformation monitoring. Fibre-optic cables are relatively inexpensive, require little to no maintenance, and can be deployed in environments that were previously impractical for or inaccessible to

traditional seismometers, such as urban environments (Dou et al., 2017; Fang et al., 2020), glaciers and permafrost regions (Ajo-Franklin et al., 2017; Walter et al., 2020), deep boreholes (Cole et al., 2018; Lellouch et al., 2019), and in lakes and submarine environments (Lindsey et al., 2019; Sladen et al., 2019) – see also Zhan (2020) for a concise review of applications in geosciences. DAS thus has an enormous potential to complement or replace seismometer arrays (Jousset et al., 2018).

However, the measurement principles of DAS are inherently different from those of conventional seismometers (Zhan, 2020), which presents new challenges in interpreting DAS data. Traditional array processing techniques, such as seismic beamforming, need to be re-evaluated for the application to DAS data.

Even though several studies already reported first results on applying seismic beamforming to linear and L-shaped DAS arrays (Fang et al., 2020; Lindsey et al., 2017, 2019), the potential of DAS in beamforming requires further exploration. In this study, we directly compare beamforming results of data from a nodal seismometer array and from a co-located optical fibre cable at the Brady Hot Springs geothermal site, Nevada, USA (Feigl and the PoroTomo Team, 2018). Specifically, we analyse the recordings of the March 2016 $M_L$ 4.3 Hawthorne earthquake, which occurred 150 km south of the Brady Hot Springs site and was well captured by both the nodal and DAS arrays. The comparison suggests that the beamforming of the DAS-recorded waveforms is severely hampered by shallow seismic scattering and by spatial variations in phase velocities, to which DAS measurements are highly sensitive. This is consistent with previous theoretical findings that ground motion gradients (or strains) are more severely affected by small-scale heterogeneities than ground motions themselves. To remedy this, we propose a method in which we convert the DAS strain rates to particle velocities, yielding DAS beamforming results that are on par with those of the nodal array. We conclude by putting these observations in a broader context of beamforming capabilities of DAS arrays of larger aperture, and their application in seismic source monitoring and earthquake early warning.

## 2   Methods

### 2.1   The PoroTomo experiment

The Poroelastic Tomography (PoroTomo) project is a hydrogeological experiment conducted in March 2016 (phase II) at a geothermal site near Brady Hot Springs, Nevada, USA (Feigl and the PoroTomo Team, 2018) – see Fig. 1. For the purpose of high-resolution monitoring of changes in rock-mechanical properties during operation of the enhanced geothermal system, an array of 238 Fairfield Nodal ZLand 3C seismometers was deployed over an area spanning 1500 by 500 m, as well as several fibre-optic cables for Distributed Acoustic Sensing and Distributed Temperature Sensing. These fibre-optic cables were laid-out horizontally in a trench of 8700 m in total length and 0.5 m in depth, and vertically in a borehole down to 400 m. The gauge length was taken to be 10 m, which was supersampled to give a channel spacing of 1 m (i.e. one strain rate measurement was made every 1 m). The geothermal reservoir of Tertiary volcanic rocks is overlain by a thick alluvium of several hundreds of metres in thickness (Jolie et al., 2015). The near-surface velocity structure of the site has been inferred from the analysis of high-frequency vibroseis sweeps and from Noise Correlation Functions (Feigl and the PoroTomo Team, 2018), showing strong variations over distances of tens of metres.

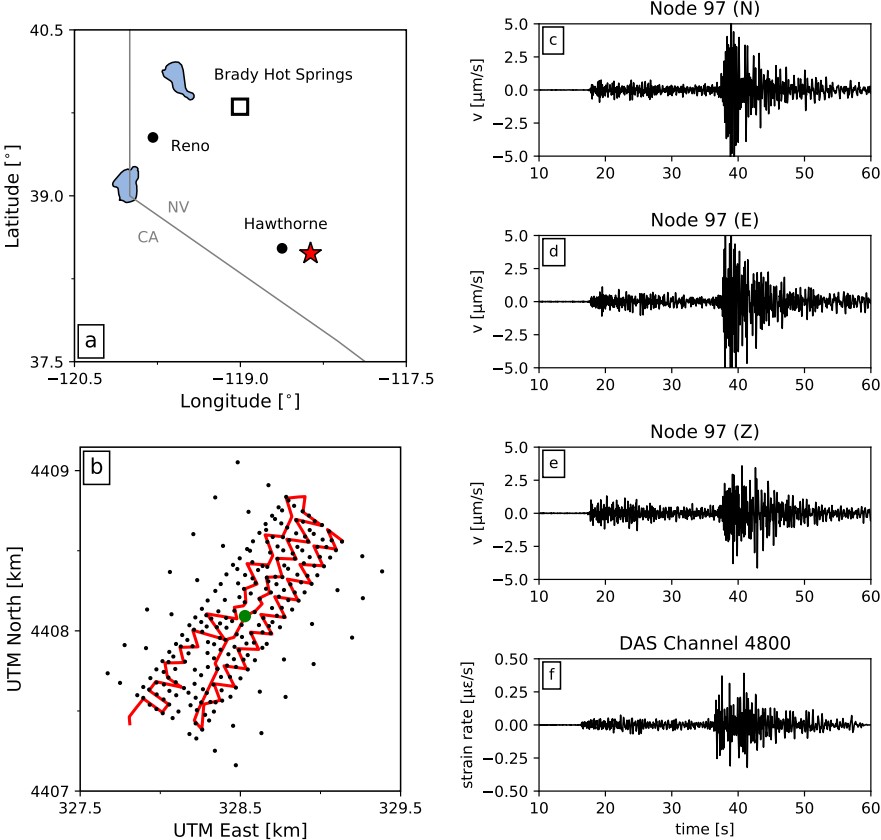

**Figure 1.** a) Location of the Brady Hot Springs natural laboratory and the March 2016 $M_L$ 4.3 Hawthorne earthquake epicentre; b) Layout of the PoroTomo nodal seismometer array (black dots) and the fibre-optic cable (red line); c-e) Ground motions recorded on the N/E/Z-components of nodal seismometer #97, which is marked in panel b by a green dot; f) Strain rates recorded by a channel co-located with nodal seismometer #97. All waveforms are filtered in a 0.5-10 Hz pass band.

During the experiment, on 21 March 2016 at 07:37:10 UTC, an $M_L$ 4.3 strike-slip earthquake occurred 150 km SSE of the geothermal site at a depth of 9.9 km. The ground motion data of this event recorded by both the nodal and DAS arrays

are available at the National Geothermal Data Repository (Feigl, 2016a, b). For convenience, we downsample the nodal and DAS data to 100 Hz, which is largely sufficient for the frequency bands selected in our analysis (up to 5 Hz). The nodal data is corrected for the geophone instrument response (damping factor of 0.7 at a 5 Hz corner frequency). An in-depth analysis of the ground motions in terms of frequency content, spatial variability of signal-to-noise ratios, and the comparison between DAS and nodal seismometers was performed by Wang et al. (2018). These authors also performed a preliminary beamforming

analysis using the data from the nodal seismmometer array, but did not attempt to make a comparison with DAS data. In the

present study, we retrieve and interpret the same data set as was analysed by Wang et al. (2018), and so we build upon the conclusions drawn from this previous study.

## 2.2 MUSIC beamforming

Seismic beamforming is a commonly employed array processing technique for estimating the direction-of-arrival (azimuth) and slowness of the seismic waves arriving at a seismic array (Capon et al., 1967; Hutchison and Ghosh, 2017; Krüger et al., 1993). It is assumed in most beamforming applications that the signal recorded at the $k$-th station in the array can be represented by a superposition of $N$ plane waves, each carrying a signal $s$ and impinging on the array at an angle $\theta$. We consider arrays deployed at the surface, thus $\theta$ is the azimuth of propagation of the incident wave. Throughout the study we assume a single source ($N = 1$), so that the frequency-domain representation of the recorded signal can be written as:

$$x_k(\omega) = a_k(\omega, S, \theta)s(\omega) + e_k(\omega) \tag{1}$$

where $e_k$ is the noise recorded at the $k$-th station, and $a_k = e^{i\omega\tau_k}$ is the steering vector that dictates the phase shift (time delay) of the signals at each station, relative to the centre of the array. The theoretical time delay $\tau_k = -S\left(\Delta x_k \sin\theta + \Delta y_k \cos\theta\right)$ is computed over a grid of candidate apparent slowness values $S$ and azimuths $\theta$, with a given station location $(\Delta x_k, \Delta y_k)$ relative to the centre of the array. In traditional delay-and-sum beamforming, the likelihood of each candidate in the grid of $S$ and $\theta$ is estimated as the projection of the steering vector $a$ onto the covariance matrix $C^2$, defined as:

$$C_{ij}^2 = \frac{x_i \bar{x}_j}{\sqrt{|x_i|^2 |x_j|^2}} \tag{2}$$

where $\bar{x}$ denotes the complex conjugate of $x$. Here, the spectra and cross-spectra involved in the equation above are estimated by the multi-taper method (Thomson, 1982), following Meng et al. (2011). Note that the covariance matrix is complex, and that it is scaled by the norms of the waveforms $x$ such that $0 \leq |C^2| \leq 1$. Consequently, the magnitude of $C^2$ is not affected by amplitude differences between $x_i$ and $x_j$, e.g. due to spatial variations in coupling or fibre orientation, which could be represented as a station-specific factor $\alpha_k(\omega)$ multiplying the first term of the right-hand-side of Eq. (1). However, local effects leading to spatial variability of waveform shape are not compensated by this normalisation.

MUltiple SIgnal Classification (MUSIC) is an extension of classical beamforming approaches that acknowledges sparsity in the number of signals arriving at the array, resulting in higher-resolution estimates of the back-azimuth and slowness of the seismic waves (Goldstein and Archuleta, 1987; Meng et al., 2011; Schmidt, 1986). Instead of projecting the steering vectors onto the full covariance matrix, a pseudo-power of the signal is estimated as the reciprocal of the projection of the steering vectors onto the noise-space of the covariance matrix, which is found through an eigenvalue decomposition of $C^2$. The procedure of estimating $C^2$ is as described above, and so the sole difference between MUSIC and traditional beamforming lies in the projection of the steering vectors onto the noise space (and taking the reciprocal), rather than projecting onto the full space of $C^2$. For a detailed exposition of MUSIC, the reader is referred to Schmidt (1986).

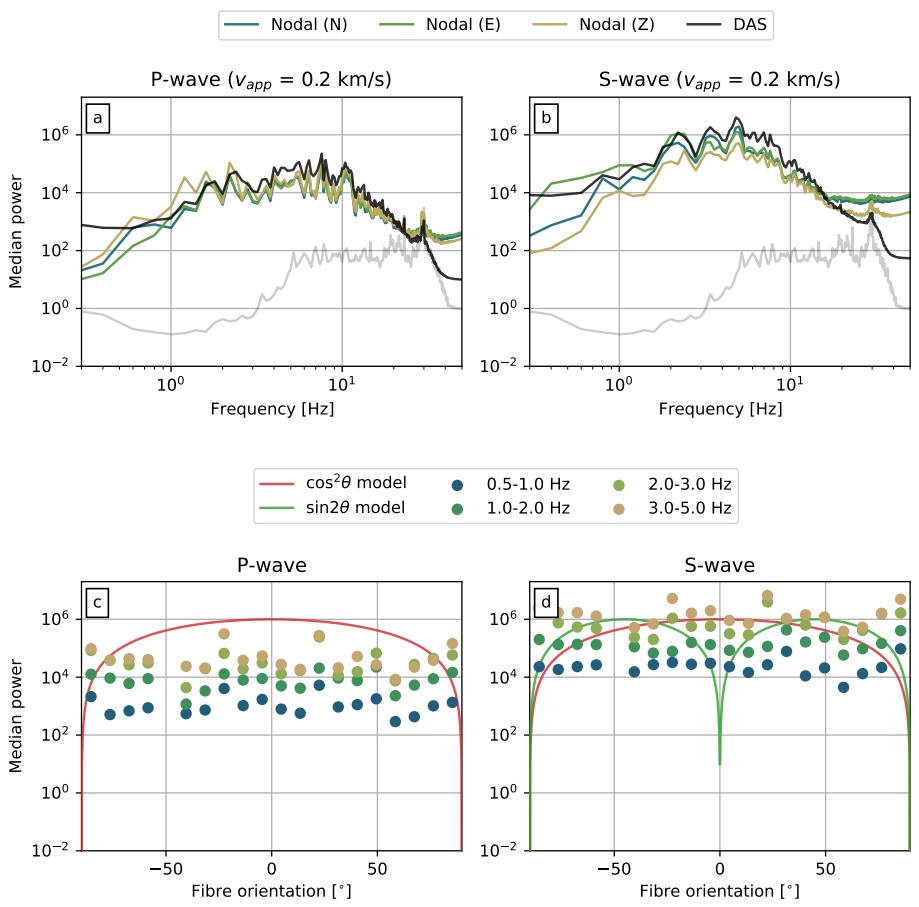

**Figure 2.** a-b) Median power spectral densities of the nodal and DAS arrays for the P-wave and S-wave. The nodal seismometer recordings are converted into acceleration spectra, which are proportional to the DAS strain rate spectra. The proportionality constant (the apparent phase velocity) is taken as $0.2 \, \text{km s}^{-1}$. The noise floor is shown as a grey band; c-d) Mean power in selected frequency bins as a function of the fibre-optic cable orientation with respect to the back-azimuth of the seismic source. The theoretical sensitivities (Martin et al., 2018) are included for reference.

## 3  Results

### 3.1  Signal characteristics and coherence

Before attempting to perform beamforming on the array data, we first consider the spectral characteristics of the recorded signals – see Fig. 2a and b. The velocity spectra of the three components of the wavefield recorded by the nodal array are first converted into acceleration spectra, which are proportional to the DAS strain rate spectra under the assumption of a single plane wave, with the phase velocity as the proportionality constant. An apparent phase velocity of $0.2 \, \text{km s}^{-1}$ for both the P-

and S-phases gives a good comparison between the nodal spectra and the DAS spectra, which suggests that a common type of wave (e.g. scattered surface waves) dominates the spectra of these time windows. For reference, the median power of the noise recorded by the nodal array prior to the P-wave arrivals is indicated by a grey line in Fig. 2. The durations of the noise, P-, and S-wave windows over which the spectral power was computed are all taken to be $5$ s, which is long enough to include low frequency information.

Owing to the nature of the measurement principle of DAS (i.e. measuring strains rather than particle motions), the directional sensitivity of the fibre to P- and S-waves is different from nodal seismometers (Kuvshinov, 2016; Zhan, 2020). For a gauge length that is much smaller than the seismic wavelength, the DAS strain rate is proportional to $\cos^2 \theta$ for a P-wave or SV-wave, and $\sin 2\theta$ for an SH-wave, assuming a plane wave with incidence angle $\theta$ relative to the fibre (Martin et al., 2018). These theoretical sensitivities are plotted for reference in Fig. 2c and d, alongside the mean power measured within selected fibre orientation bins. As was also concluded by Wang et al. (2018) from analysing the directional dependence of the signal-to-noise ratio, no directionality of the mean power is observed. Moreover, the variability within a given frequency band exceeds one order of magnitude. Wang et al. (2018) interpreted this as an effect of the heterogeneous site response, which likely exerts a first-order control on the amplitudes and directionality of the ground motions. This will be demonstrated in more detail in Section 3.4.

### 3.2 Beamforming results of the nodal and DAS arrays

To set a baseline, we first beamform the P- and S-waves recorded by the nodal array for each component separately. We take a time window from $2$ s before to $8$ s after the first arrival of each respective phase (i.e. $10$ s in total). To visualise the coherence of the wavefield in each direction, we select a 10-second time window starting near the P-arrival. The waveforms are then ordered by distance from the earthquake epicentre, band-pass filtered in the 0.5-1 Hz range, and each trace is scaled by its standard deviation – see Fig. 3. In particular the vertical waveforms exhibit very strong coherence across the entire array. Among the horizontal components, the N-component is more coherent, consistent with a source that is oriented almost directly south of the array (with a back-azimuth of $157°$ from the centre of the array). Similarly, the S-waves (not shown here) exhibit strong coherence particularly in the E-direction, followed by the N- and Z-directions.

The P-wave beamforming results using all the nodes in the nodal array show a well-resolved source in the southeast, with an azimuth close to the true back-azimuth of $157°$ (Fig. 4). As expected from the waveform coherence, this source is most stable and well-resolved for the vertical component, with an apparent propagation velocity between 4 and 6 km s$^{-1}$. Only in the 0.5-1.0 Hz frequency band does the beamforming of this component lead to a relatively poorly resolved location, which may be due to the influence of the corner frequency (typically around 1-3 Hz for an $M_L$ 4.3 event; (see e.g. Scholz, 2019)). The beams formed from the N-component also indicate a southeast direction-of-arrival, but with a less well-resolved apparent velocity. The beams formed from the E-component suggest weak, poorly resolved sources in the south, west, and east, which are likely scattered P-waves. The sources indicated by beamforming of the S-wave (Fig. 5) are even better resolved than the P-wave sources, particularly in the E-component, with an apparent propagation velocity between 2 and 4 km s$^{-1}$. For an assumed true P- and S-wave speeds of 2.1 and 1.3 km s$^{-1}$, respectively (crudely estimated from Feigl and the PoroTomo Team, 2018), the

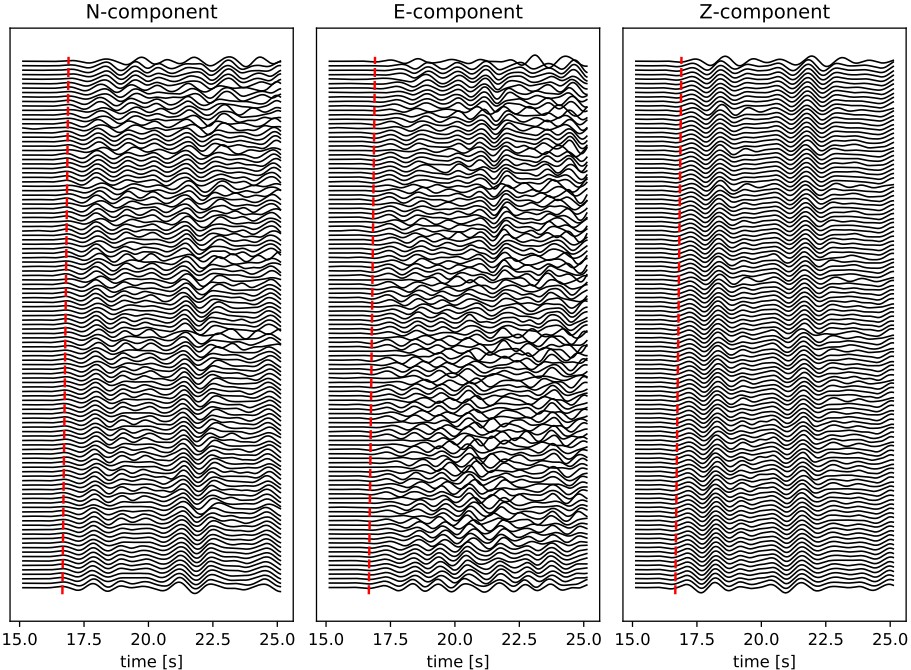

**Figure 3.** P-waveforms recorded by the nodal array, ordered by distance from the earthquake epicentre, band-pass filtered in the 0.5-1 Hz range, and scaled by the standard deviation of each trace. Time is relative to the start of the recordings, and is synchronised with the waveforms shown in Fig. 1. For reference, the move-out is indicated by the red dashed line.

inferred apparent velocities would correspond with an inclination of the direction-of-arrival of 65° (consistent with the ratio of vertical to horizontal amplitudes of the nodal P-waveforms measured across the nodal array).

In strong contrast to the nodal array, the P- and S-waveforms recorded by the DAS array show a low degree of coherence (Fig. 6). While individual cable segments may exhibit some internal coherence (analysed further in Section 3.4), this coherence does not persist across the array. There are several factors that contribute to the incoherence of the recorded signals. Firstly, for a horizontal cable, the DAS strain is a combination of gradients of the two horizontal components of the wavefield, which may lead to unfavourable interference. Secondly, the amplitudes of the DAS recordings depend strongly on the coupling of the fibre-optic cable to the ground (Wang et al., 2018) and on the angle of incidence of the incoming plane wave (Martin et al., 2018), so that various segments at different locations and with different fibre orientations experience variable signal-to-noise ratios. Thirdly, depending on the orientation of the fibre, S-wave polarity flips are anticipated (Fang et al., 2020). These polarity flips are due to the projection of the particle motion onto the fibre, leading to contraction in some segments and extension in others (Lindsey et al., 2017). Lastly, spatial gradients of the particle velocity (i.e. strain rates) are highly sensitive to local heterogeneities (Singh et al., 2020), and their amplitudes are inversely proportional to the apparent phase velocity so that slow waves (often scattered waves) are amplified.

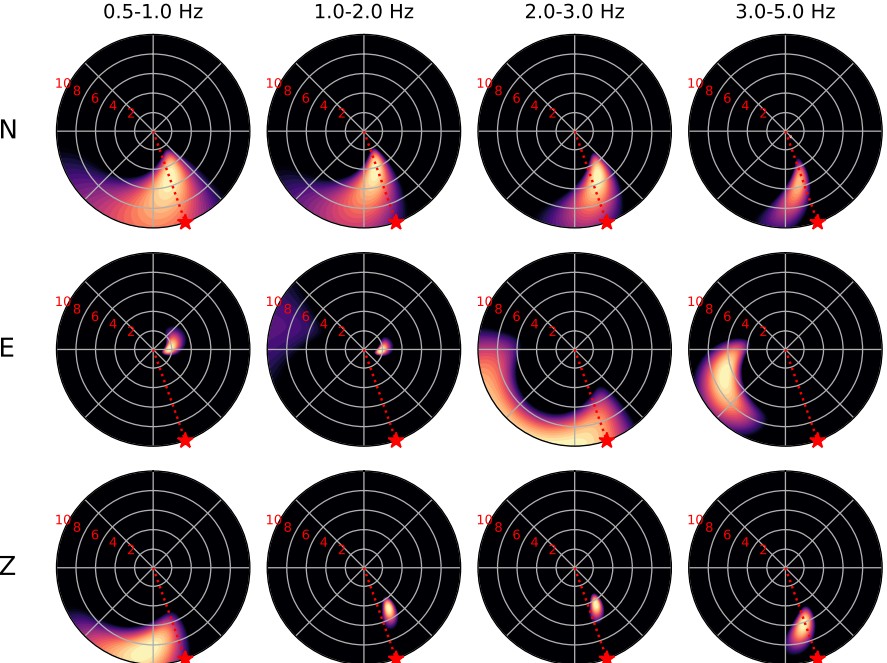

**Figure 4.** Beamforming results of the P-waves recorded by the nodal array. Each panel shows the MUSIC pseudo-power for the candidate combination of azimuth and apparent velocity (reciprocal of slowness) normalised to lie between 0 and 1. For visual clarity we clip the colours at a normalised pseudo-power of 0.8 (i.e. at 80 % of the maximum pseudo-power). The apparent velocity is plotted up to $10~\mathrm{km\,s^{-1}}$, with radial grid increments of $2~\mathrm{km\,s^{-1}}$ (indicated by the red numbers). The frequency bands and components are indicated above each column and beside each row, respectively. The true back-azimuth of the source is indicated by a red star in each panel.

When we nonetheless continue to perform beamforming on the entire DAS array recordings (8621 channels in total), we obtain highly variable results (Fig. 7) for the same window length and frequency range as was used for the nodal array. At the lower frequencies (below $2~\mathrm{Hz}$), we find a diffuse spread of pseudo-power over a range of potential source azimuths and apparent velocities. By contrast, at the higher frequencies, we find several well-resolved sources pointing in the southeast and east directions, but with very low apparent velocities (less than $2~\mathrm{km\,s^{-1}}$). These apparent velocities between 1 and $2~\mathrm{km\,s^{-1}}$

are consistent with the inferred S-wave speeds at depths of a few hundred metres, suggesting a shallow "source" (most likely a seismic scatterer). As mentioned above, slow phase velocities amplify the recorded DAS strain rates, and so it is not unexpected that, in the absence of strong coherent direct arrivals, slow, scattered waves dominate the beamforming solutions. Also recall that through the definition of the covariance matrix, the absolute amplitude of the recorded signals is irrelevant, which partially addresses the issues of (potential) directionality and ground coupling. However, the wavefield is composed of multiple waves

(e.g. direct and scattered arrivals) and their relative phase amplitudes may vary with fibre orientation, which still affects the pseudo-power.

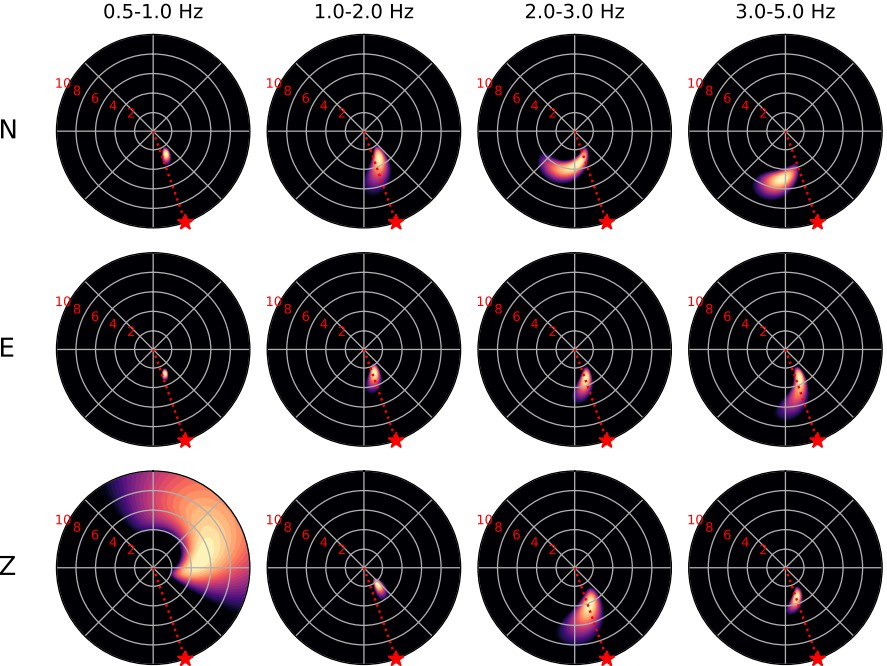

**Figure 5.** Beamforming results of the S-waves recorded by the nodal array. The panel representation is the same as in Fig. 4.

### 3.3 Simulating DAS recordings from the nodal array

In the previous section, we pointed out several potential reasons for the lack of waveform coherence and the inconsistent beamforming results. In the following section, we will explore one of these factors that is inherent to the DAS measurement

principle: the one-dimensional strain rate measurement that aggregates multiple components of the particle velocity field. A DAS measurement provides only one component of strain, the longitudinal strain along the direction of the fibre. This limitation can be mitigated by making the measurements along helically-wound cables (Kuvshinov, 2016), but since such cable designs were not deployed during the PoroTomo experiment, one has to resort to alternative approaches.

As has been clearly demonstrated by Wang et al. (2018), the particle velocity measurements recorded on two nodal seis-

170 mometers separated by a small distance $L$ can be accurately converted into the average longitudinal strain rate $\dot{\varepsilon}$ between the two nodes (expressed here at their midpoint $x$):

$$\dot{\varepsilon}(x) = \frac{1}{L}\left[\dot{u}\left(x + \frac{L}{2}\right) - \dot{u}\left(x - \frac{L}{2}\right)\right] \tag{3}$$

where $\dot{u}$ is the particle velocity in the direction parallel to the positional difference vector between the two nodes. This average strain rate is equal (or proportional) to the DAS strain measured along a gauge length $L$ whose end points are co-located with

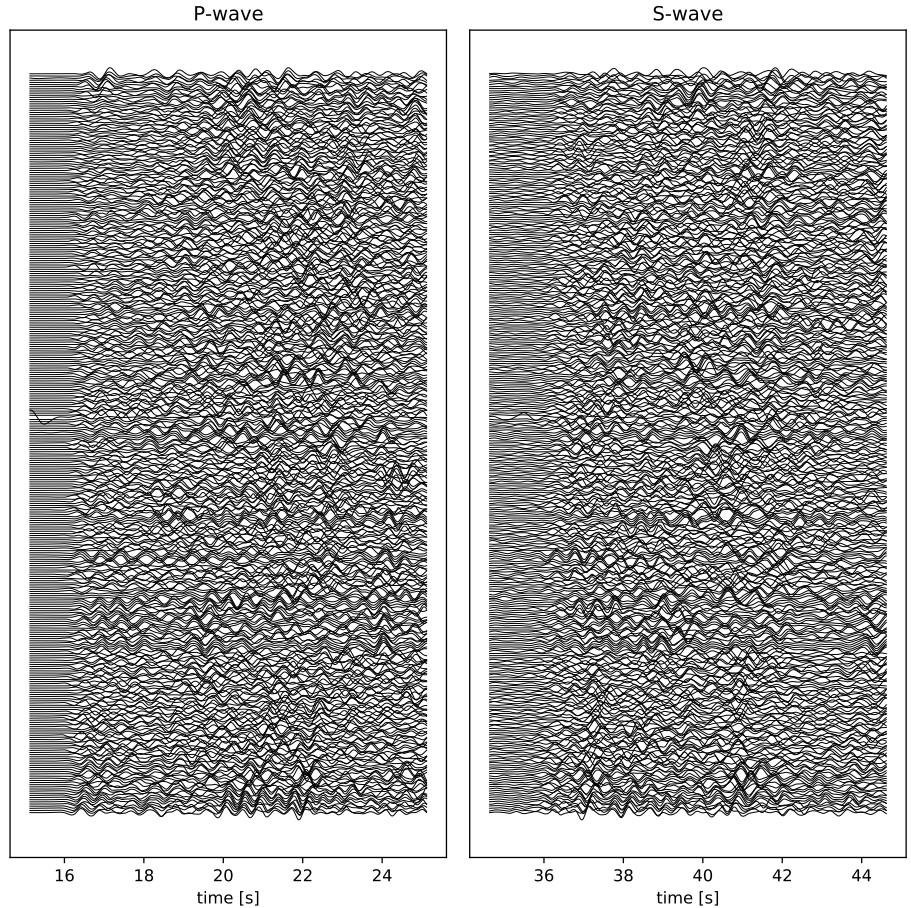

**Figure 6.** P- and S-waveforms recorded by the DAS array, ordered by distance from the earthquake epicentre, filtered in a 0.5-2 Hz pass band, and scaled by the standard deviation of each trace. Time is relative to the start of the recordings, and is synchronised with the waveforms shown in Fig. 1. For clarity, only every $20^{\text{th}}$ channel waveform is plotted.

the nodes, if the cable is straight and has a spatially uniform coupling between the two nodes. A similar relation holds when the distance between the nodes is a multiple of the DAS gauge length (see Wang et al., 2018, their Eq. 5).

Given the density of the nodal array, in which most nodes are positioned less than $100$ m from their nearest neighbour, we can use this relationship to simulate the response of a DAS array to the strain field induced by the Hawthorne earthquake, and test the effect of superimposing multiple independent components on the beamforming performance. To this end, we triangulate the

180 node coordinates (Fig. 8a), yielding pairs of stations that define each edge in the mesh. For node pairs separated by a distance less than $80$ m, we rotate the horizontal (N and E) components of velocity onto the 'virtual' DAS fibre orientation $\theta$ (relative to east) as $\dot{u} = \dot{u}^{\text{N}} \sin\theta + \dot{u}^{\text{E}} \cos\theta$. Substitution of $\dot{u}$ into Eq. (3) yields the mean strain rate along the simulated DAS fibre in

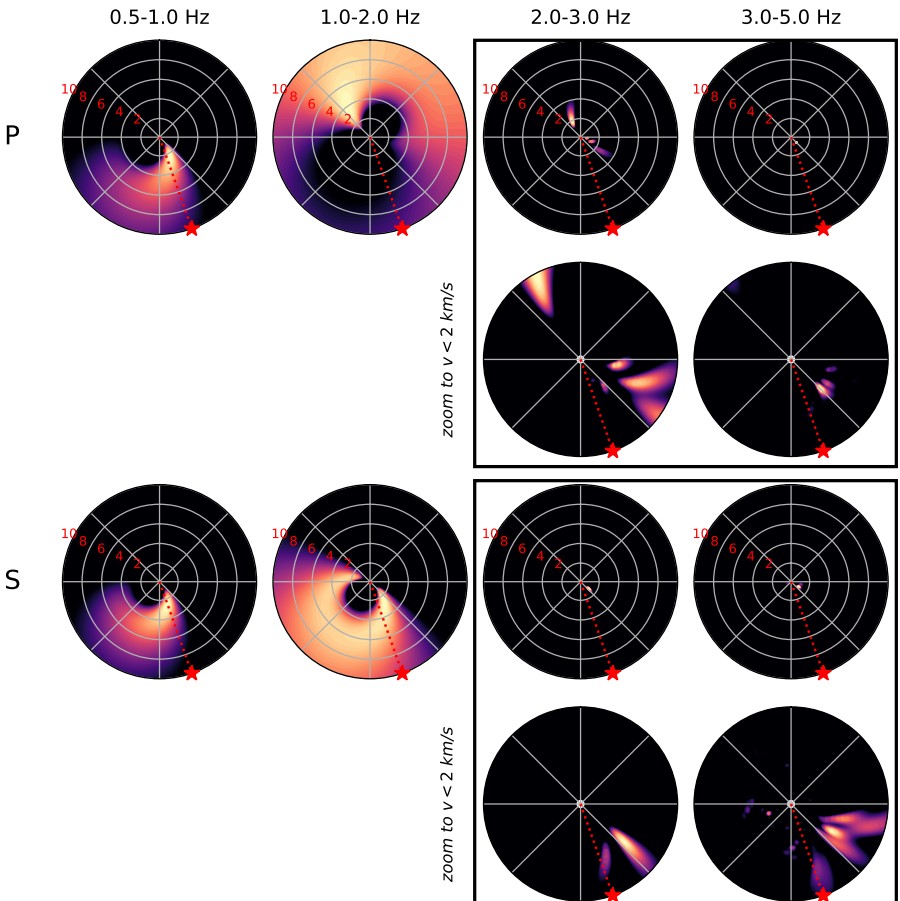

**Figure 7.** Beamforming results of the P- and S-waves recorded by the DAS array. The panel representation is the same as in Fig. 4. To be able to visually resolve the slow sources at frequencies greater than 2 Hz, we add additional panels magnifying the region up to an apparent velocity of $2 \text{ km s}^{-1}$.

between each pair of stations (414 in total):

$$\dot{\varepsilon}\left(\frac{x_A + x_B}{2}, \frac{y_A + y_B}{2}\right) = \frac{\left(\dot{u}_A^N - \dot{u}_B^N\right)\sin\theta + \left(\dot{u}_A^E - \dot{u}_B^E\right)\cos\theta}{\sqrt{\left(x_A - x_B\right)^2 + \left(y_A - y_B\right)^2}} \tag{4}$$

where the subscripts $A$ and $B$ indicate the two seismometers between which the strain rate is calculated, each located at a coordinate point $(x, y)$.

The resulting simulated strain rate P-waveforms are shown in Fig. 8b, for selected segments with an orientation within $\pm 10°$ from the event back-azimuth (red segments in Fig. 8a). Even though individually the N- and E-components recorded by the nodal stations exhibit some coherence across the array (see Fig. 3), the horizontal strain rates, involving differences of

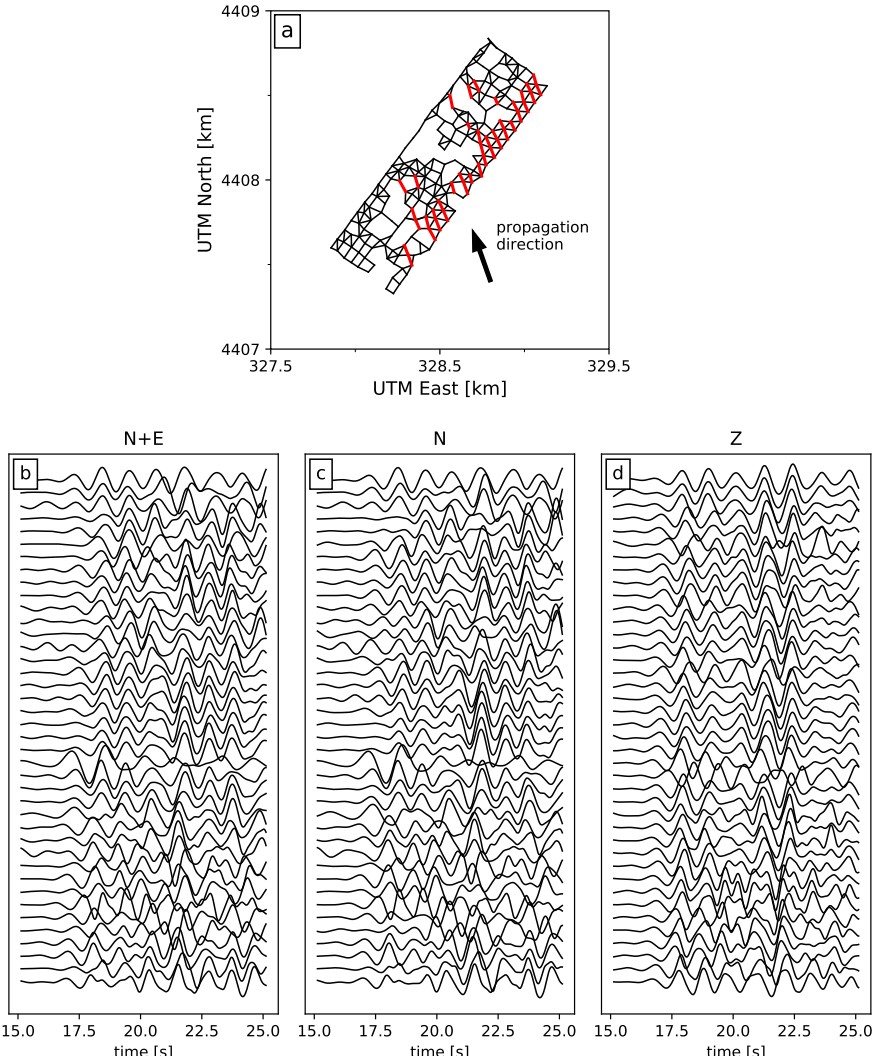

**Figure 8.** a) Layout of the virtual DAS array, defined by the edges connecting two nodal seismometers separated by less than $80$ m in distance. The propagation direction of the wavefield is included for reference; b-d) P-waveforms recorded by the virtual DAS array for segments with an orientation $\pm10°$ with respect to the back-azimuth of the seismic source (indicated in red in a)), ordered by distance from the seismic source, filtered in a $0.5$-$1$ Hz pass band, and scaled by the standard deviation of each trace. In b), the waveforms are a superposition of the N- and E-components of the nodal seismometers, while in c) and d) only the N- and Z-components are used, respectively.

velocities on two horizontal components, are not coherent. Moreover, if the strain rate (the gradient of the particle velocity field) is calculated only on the basis of the strongly coherent N- or Z-components (Fig. 8c and d, respectively), then the coherence that is seen in the particle velocity measurements (Fig. 3) is almost completely lost. This indicates that it is not

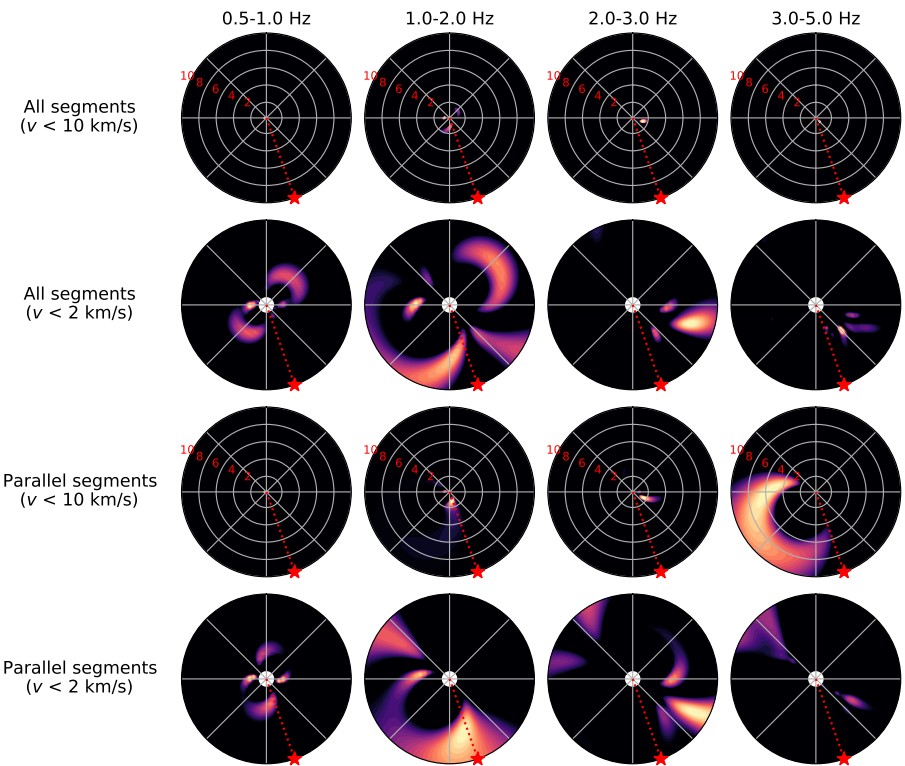

**Figure 9.** Beamforming results of the P-waveforms recorded by the virtual DAS array, derived from the E- and N-components of the nodal seismometers (i.e. Eq. (4)). The panel representation is the same as in Fig. 4. Each second row of panels is a magnification of the upper row, highlighting the sources with low apparent phase velocity ($< 2 \, \mathrm{km \, s^{-1}}$). The upper two rows are the results of beamforming all the segments, while the lower rows are the results of beamforming only sub-parallel segments (indicated in red in Fig. 8a).

just the superposition of two components that causes destructive interference, but that this is caused by the gradient operator itself (regardless whether this operation be done mathematically, like was done here, or physically, like in a DAS fibre). Also, since only segments were selected that are near-parallel to each other, the lack of waveform coherence cannot be attributed to directionality effects.

When we perform the beamforming on the P-waveforms recorded by the virtual DAS array (all segments; Fig. 9), the only sources that stand out are those with very slow apparent phase velocity ($< 2 \, \mathrm{km \, s^{-1}}$) and azimuths that vary from west to east. Since the overall waveform coherence across the array is low, these sources likely result from subregions in the array that locally exhibit moderate coherence, but which does not persist throughout the array. Owing to the directional sensitivity of the (simulated) DAS measurement, combining segments of different orientations may affect the beamforming results. We repeated the beamforming on selected segments with an orientation $\pm 10°$ from the event back-azimuth (see Fig. 8a). When only these

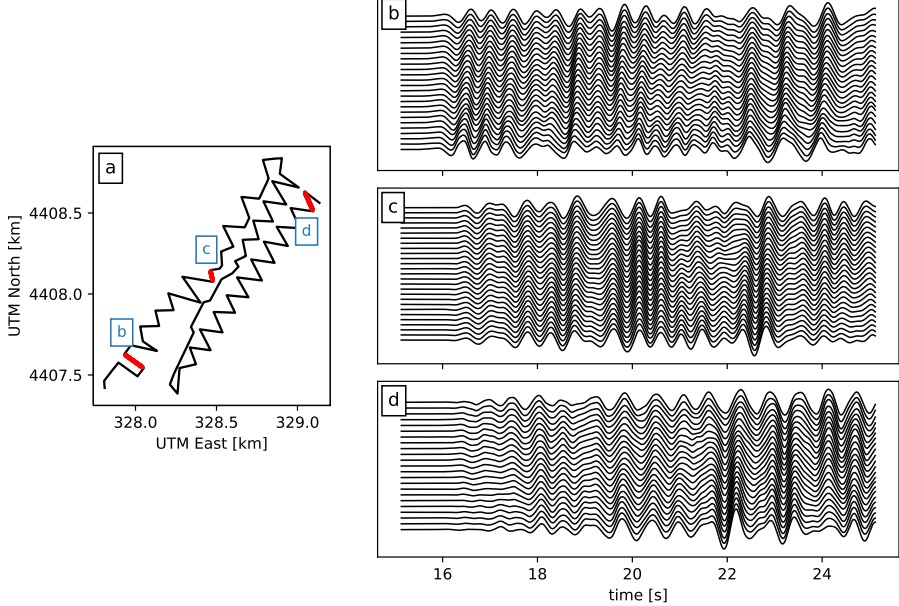

**Figure 10.** Location of the selected DAS cable segments, indicated in red in panel a. The corresponding P-waveforms that are shown in panels b, c, and d are filtered in a 0.5-2 Hz pass band and scaled by the standard deviation of each trace.

sub-parallel segments are selected, an ambiguous source arises in the west with high apparent velocity, which is inconsistent with the back-azimuth and apparent phase velocities of the seismic source. This exercise demonstrates that the measurement

principle of DAS challenges beamforming methods that are traditionally applied to particle velocities rather than to strain rates. Since we derived the strain rates directly from the nodal seismometers, the lack of coherence and beam resolution seen in the DAS data (Figs. 6 and 7) cannot be attributed to DAS-specific technicalities like coupling of the DAS cable with the ground or phase unwrapping artefacts, since the nodal seismometers and their derived data do not suffer from this.

### 3.4    Selective beamforming of the DAS array

Even though the DAS array as a whole does not exhibit strong waveform coherence, there are short individual segments that do exhibit excellent coherence locally. Examples of this can be found in Wang et al. (2018) (their Fig. 14), who selected three segments to estimate the apparent P-wave speed from the first P-wave arrivals recorded by the DAS fibre. The apparent phase velocities obtained from this analysis ranged from $1.124$ to $1.452 \, \mathrm{km \, s^{-1}}$, which are much lower than the apparent velocities obtained from the nodal array beamforming (between 4 and $6 \, \mathrm{km \, s^{-1}}$), and are suggestive of a shallow, scattered source.

Since these segments exhibit strong waveform coherence (Fig. 10), we can attempt to form a stable beam by selecting only the channels associated with these segments.

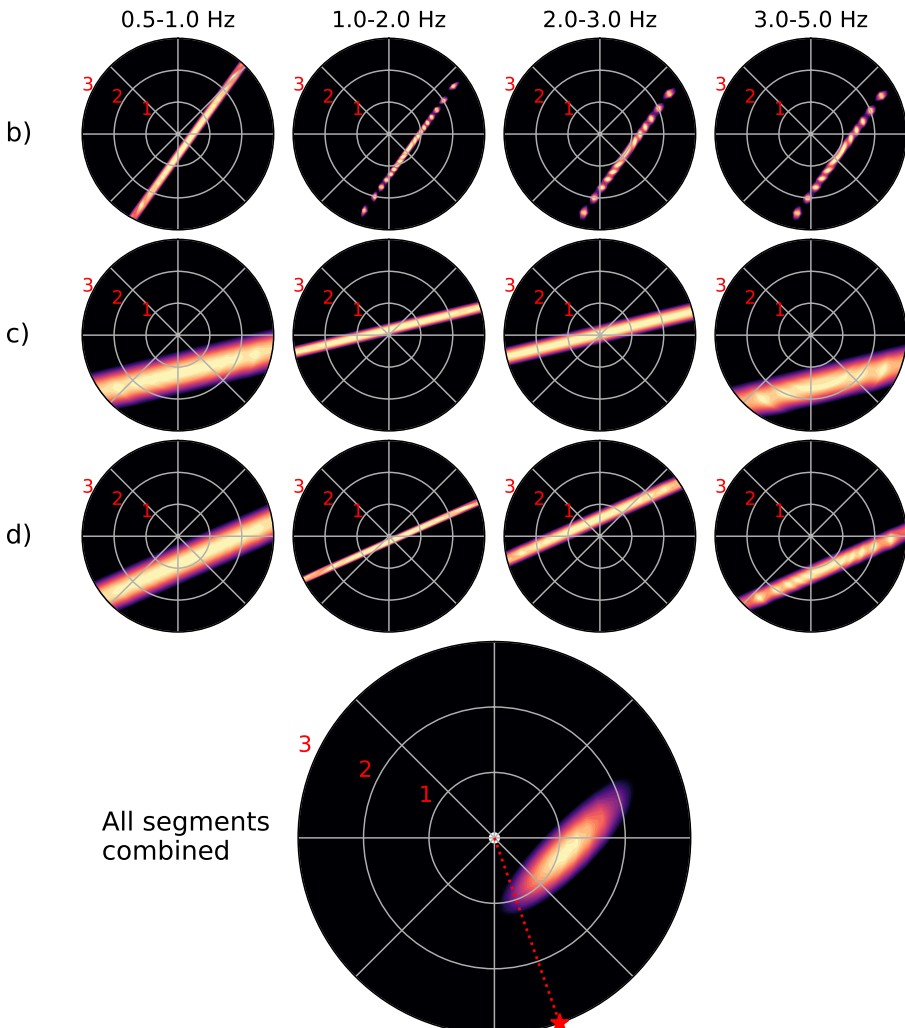

**Figure 11.** Beamforming results of the P-waveforms recorded on selected segments of the DAS array. In this figure, the panels show pseudo-power $\hat{P}$ normalised by the maximum power, such that $0 \leq \hat{P} \leq 1$, plotted as a function of the back-azimuth and apparent slowness of the candidate source. The slowness values (in $\mathrm{s\,km^{-1}}$) of the radial grid lines are as indicated by the red numbers. The letters indicated on the left of each row correspond to the panels in Fig. 10.

Beamforming on multiple small, linear segments comes with additional challenges. First of all, even though the waveforms may be coherent within one segment, they are not necessarily coherent between segments, which affects the beamforming performance when the waveforms of all the segments are combined. On the other hand, if beamforming is performed on each segment individually, ambiguity arises from the linear geometry of the segments. A plane wave travelling parallel to the length of the cable at a high speed will induce the same phase shift as a plane wave impinging on the cable with a larger angle but at

a lower speed. In other words, there exists a perfect trade-off between back-azimuth and apparent velocity for linear segments. This trade-off is most clearly seen when plotting the beam pseudo-power in a space spanned by the azimuth and apparent slowness (reciprocal apparent velocity), in which the beam pseudo-power will appear as a straight line perpendicular to the orientation of the cable (see Fig. 11; see also Fig. 2e in Lindsey et al. (2019)). This ambiguity can be resolved by combining the beamforming results of multiple segments: the source azimuth and slowness that produces the phase shifts consistent with all segments is the one where the linear bands of beam power intersect. As laid out by Rieken and Fuhrmann (2004), the intersection of the invididual signal spaces of all subarrays can be found by the method of projection onto convex sets. From this analysis, it follows that the signal space of $M$ subarrays combined furnish a block matrix, which upon substitution into the definition of the MUSIC spectrum yields (Rieken and Fuhrmann, 2004):

$$\hat{P} = \left( \sum_{m=1}^{M} \mathbf{a}_m^H \mathbf{G}_m \mathbf{G}_m^H \mathbf{a}_m \right)^{-1} = \left( \sum_{m=1}^{M} \hat{P}_m^{-1} \right)^{-1} \tag{5}$$

with $\mathbf{G}_m$ representing the projection onto the noise-subspace of the covariance matrix $\mathbf{C}_m^2$ (as defined in Eq. (2)) and $\mathbf{a}_m$ the steering vector of the $m$-th subarray, and the superscript $H$ denoting the conjugate transpose. In other words, the analysis of the intersections of signal subspaces of the subarrays naturally leads to a harmonic mean of the MUSIC pseudo-spectra of each subarray. When applying this rationale to the subarray beams presented in Fig. 11, we obtain a combined beam which shows a well-resolved source with a back-azimuth towards the east and an apparent slowness of $1.2 \text{ s km}^{-1}$ ($0.83 \text{ km s}^{-1}$). Consistent with our previous interpretations, the azimuth and phase velocity of this source are incompatible with direct P-wave arrivals from the Hawthorne earthquake; it is therefore interpreted as a shallow scattered wave.

Since the internal waveform coherence of each segment varies across the array (and also with the selected frequency band), manually selecting a few segments for the beamforming may introduce a bias. However, if internal waveform coherence is the main selection criterion, whether or not to include individual segments in the beamforming analysis can be determined on the basis of the $L_2$-norm of the covariance matrix, i.e.:

$$c^2 = \frac{1}{N^2} \sum_i \sum_j C_{ij} \bar{C}_{ij} \tag{6}$$

where $N$ is the number of channels in a given segment for which the covariance matrix $C_{ij}$ is computed. We compute $c^2$ for the P-waveforms of each quasi-linear segment of the cable over the 0.5-1 Hz frequency band, and select those segments with $c^2 > 0.9$ (six in total; Fig. 12a). Since these segments are all linear, we obtain ambiguous results in terms of the azimuth and apparent velocity (which trade-off with one another), but within this ambiguity the sources are well resolved (Fig. 12b). However, while segments 1,2, 3, and 5 suggest a direction-of-arrival between the east and the south with a maximum apparent velocity of 2-3 $\text{km s}^{-1}$, the other two segments (4 and 6) suggest a direction-of-arrival from the north with a maximum apparent velocity at around 1 $\text{km s}^{-1}$, which most likely signify the predominance of locally scattered waves. When we combine the beam pseudo-power of each segment through Eq. (5), we obtain an apparent source with a back-azimuth pointing northeast, and an apparent velocity of 0.7 $\text{km s}^{-1}$ (Fig. 12c). At first, this seems counter-intuitive, as none of the segments seem to suggest a source northeast of the array, while the combined result does. Moreover, the combined apparent velocity is substantially

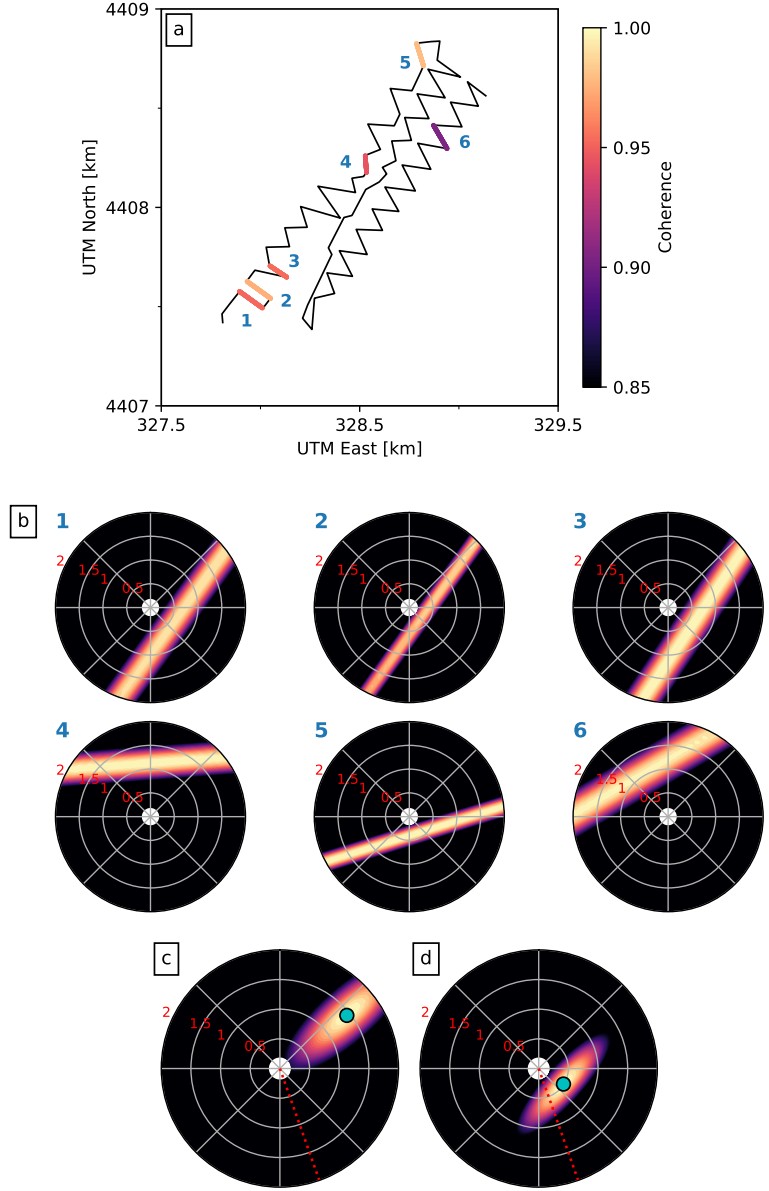

**Figure 12.** a) Locations and internal coherence of six selected segments along the cable, each indicated by a number; b) Beamforming results in the 0.5-1 Hz frequency band for each of the segments. The slowness values (in s km$^{-1}$) of the radial grid lines are as indicated by the red numbers; c) Beam pseudo-power of all the segments combined through Eq. (5); d) Beam pseudo-power of segments 1, 2, 3, and 5 combined through Eq. (5). In c) and d), the coordinate of the maximum beam power is indicated by a cyan dot.

less than the maximum apparent velocities of all the segments individually. However, these results are a direct consequence of the harmonic averaging procedure adopted here, which implicitly assumes that the wavefields at all selected subarrays are dominated by the same sets of waves: maximising the MUSIC pseudo-spectra at the intersection of the signal spaces of all the segments becomes problematic for sources that are diametrically opposite of each other (e.g. north-south), as the intersection of the signal spaces occurs only at infinity when representing the pseudo-power in slowness space (or at the origin in velocity space). This then leads to the diffuse spread of pseudo-power seen in Fig. 12c with the maximum pseudo-power at high slowness values.

When we exclude the two segments with an apparent source north of the array (so that four segments remain), we obtain a maximum beam power at an azimuth of $122°$ and apparent velocity of $2.0 \text{ km s}^{-1}$. While this result is in better agreement with the true direction of arrival of the seismic source ($157°$ with an apparent P-wave speed of $4$-$6 \text{ km s}^{-1}$, as inferred from the nodal array), the discrepancy is substantial. Particularly the low apparent velocity inferred from the DAS segments is suggestive of the arrival of scattered waves rather than direct waves. Regardless, this result is only obtained after manual quality control and selection of desired segments. For automated segment selection and beamforming one must be cautious for conflicting directions-of-arrival that lead to artificial results like in Fig. 12c.

### 3.5 Converting DAS strain rates to velocities

As demonstrated in Section 3.3, one can convert the nodal seismometer data into strain rate using Eq. (4). In doing so, much of the waveform coherence is lost, and slow phases (e.g. surface waves) are emphasised in the beamforming results. Following the same reasoning, the converse is expected to be true: to alleviate the effects of heterogeneities and slow surface waves as discussed in the previous sections, one could integrate the DAS strain rate data along the cable to obtain particle velocities. As will be shown in the following section, performing this conversion greatly improves the DAS beamforming results. However, great care must be taken in performing this analysis, and so we will start with a detailed description of the adopted procedure. We begin by taking Eq. (3) and expressing the particle velocity at a point $x$ as:

$$\dot{u}(x) = \dot{u}(x_{ref}) + \dot{\varepsilon}(x; x_{ref})\Delta x \tag{7}$$

where $\dot{u}$ is the particle velocity in the direction parallel to the fibre (defined as positive when pointing from the reference point $x_{ref}$ to $x$), $\Delta x$ is the distance between $x_{ref}$ and $x$, and $\dot{\varepsilon}(x; x_{ref})$ is the average strain rate between $x$ and $x_{ref}$ (taken positive in extension). We emphasise that $\dot{\varepsilon}$ is not a finite-difference approximation of the strain at the midpoint of the segment, but the exact average of the strain rate along the segment, and that DAS measures these strain rates averaged over one gauge length long segments. Moreover, no assumptions are made regarding the apparent propagation speed of the signals to convert strain rate into particle motion (as is typically done in other methods; e.g. Lior et al., 2021; Zhu et al., 2021, this issue). Therefore, in the case where a seismometer is co-located with the DAS cable, we can take the horizontal component of the recorded particle velocity in the direction of the fibre as a starting point, and compute the particle velocity at the next DAS channel through integration of the recorded strain rate. This procedure can then be extended to obtain the particle velocity at each subsequent

gauge length along a straight fibre by using the following equation for a segment of length $nL$ ($L$ being the gauge length):

$$\dot{u}(x_{ref} + nL) = \dot{u}(x_{ref}) + L \sum_{i=1}^{n} \dot{\varepsilon}(x_{ref} + iL; x_{ref} + (i-1)L) \tag{8}$$

Naturally this approach will accumulate integration errors as the strain rate is integrated farther and farther away from the seismometer owing to departures from the assumptions of a straight cable and uniform coupling, as well as instrument noise. We can evaluate this to some extent by considering DAS segments that have two co-located nodes, integrating from one node to the other and comparing the converted DAS particle velocity with the terminal node. It is critical to note that DAS records only one component of strain, namely the longitudinal strain associated with the local direction of the cable. Integrating the DAS recordings will therefore only yield one component of the particle motion parallel to the fibre, which cannot be decomposed into perpendicular components (like recorded by the nodal seismometers). Nevertheless, the single-direction measurement of particle velocity is likely more coherent than the single-direction measurement of strain rate.

The integration along the cable is systematically performed in the direction from low channel numbers to high channel numbers. In the current dataset, this corresponds to the direction running from the DAS interrogator to the opposite end of the cable. Let $\dot{\mathbf{u}}(\mathbf{x}_i)$ denote the 3-component ground velocity field at location $\mathbf{x}_i$ along the cable, with subscript $i$ denoting the channel index. The longitudinal (along-cable) strain rate recorded by DAS on the channel corresponding to the gauge-long segment $[\mathbf{x}_i, \mathbf{x}_{i+1}]$ is $\dot{\varepsilon}(\mathbf{x}_{i+1}, \mathbf{x}_i) = \frac{1}{L} (\dot{\mathbf{u}}(\mathbf{x}_{i+1}) - \dot{\mathbf{u}}(\mathbf{x}_i)) \cdot \mathbf{n}_i$, where $\mathbf{n}_i = \frac{1}{L} (\mathbf{x}_{i+1} - \mathbf{x}_i)$ is the unit vector pointing from $\mathbf{x}_i$ to $\mathbf{x}_{i+1}$. In the local reference frame associated with the cable, $\dot{u}$ in Eq. (8) is defined as the projection of the particle velocity in the direction of increasing channel numbers along the cable: $\dot{u} = \dot{\mathbf{u}} \cdot \mathbf{n}$. Since the cable curves and is laid out in a zig-zag pattern, even parallel cable segments may have unit vectors pointing in opposite directions. To bring all velocities to a common reference frame to be used for beamforming, after evaluating Eq. (8) for each segment we multiply $\dot{u}$ by $\text{sgn}[\mathbf{e} \cdot \mathbf{n}]$, where $\mathbf{e}$ is a reference unit vector taken here as the unit vector pointing east. If this sign correction is not performed, the integrated waveforms will change polarity depending on the orientation of the fibre, and coherence will be broken as a result.

Lastly, in the PoroTomo experiment the recordings of strain rate were made every $1$ m, i.e. the gauge length of $10$ m was oversampled by a factor 10. However, owing to the averaging effect of the gauge these additional samples are not independent of one another, and so only one independent measurement of the strain field is made every gauge length. We therefore perform the integration over non-overlapping gauges, i.e. over channels that are one gauge length away from each other, as indicated in Eq. (8).

Keeping these notations in mind, we perform the conversion of DAS strain rates to particle velocities as described above. We identify the nodal seismometers that are at most 1 gauge length ($10$ m) away from the nearest DAS channel. We then select linear portions of the DAS array that have a node at the start and at the end of the segment. For each segment we compute the horizontal component of the wavefield (parallel to the segment) as recorded by the starting node, and perform the integration along the fibre in the direction of the terminal node (accounting for the polarity as described above). For the purpose of beamforming, we select the DAS segments that are within $\pm 10°$ of east (see Fig. 13a), which should exhibit a strong sensitivity to the horizontal component of the S-wave. To demonstrate that the proposed integration method is accurate, we also select the longest quasi-linear segment in the DAS array (cyan segment in Fig. 13a) for inspection of the waveforms.

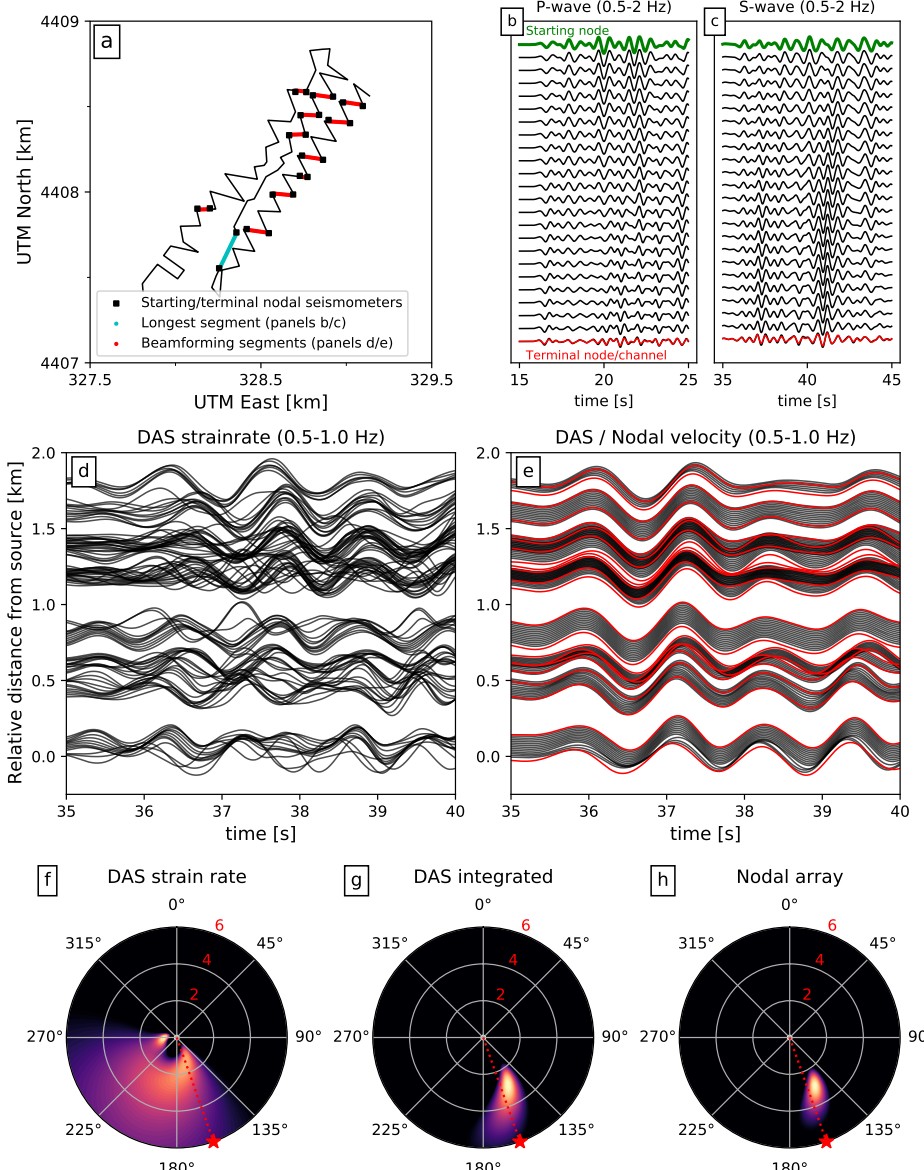

**Figure 13.** a) Locations of the east-west segments in the DAS array selected for the beamforming analysis (indicated in red) along with the starting and terminal nodal seismometers. The longest quasi-linear segment of the array is indicated in cyan.; b) Particle velocity P-waveforms resulting from the integration of the recorded strain rates, taking the nodal seismometer waveform (green) as the starting point. The terminal node waveform (red) is plotted on top of the last DAS channel for a direct comparison; c) Same as in panel b, but for the S-wave; d) DAS strain rates measured along the selected segments (see panel a); e) Particle velocities as resulting from the integration of DAS strain rates (in black), and as recorded by the starting/terminal nodal seismometers (in red); f-h) Beamforming results for the DAS strain rates, converted particle velocities, and nodal seismometer recordings, respectively. The waveforms used are as shown in panels d and e (first 5 s of the S-wave, 0.5-1 Hz frequency band). The radial increments are indicated in $\mathrm{km\,s^{-1}}$.

As can be seen in Fig. 13b and c, the particle velocity obtained from the integration of the DAS strain rates up to the last channel is practically identical to the waveforms recorded by the terminal nodal seismometer, demonstrating the accuracy of the method. The comparison is shown in Fig. 13 up to 2 Hz, but the accuracy persists up to 5 Hz. This segment is 230 m in length, and consequently the wavefield changes substantially from the start to the end of the segment (particularly for the S-wave; Fig. 13c). Nonetheless, the difference between the converted DAS- and the nodal data is minimal, which is encouraging

for the integration of DAS strain rates over even longer distances (potentially up to several kilometres).

As expected, converting the DAS strain rates to particle velocities restores the waveform coherence throughout the array (compare Fig. 13d and e). Particularly in the 0.5-1 Hz frequency band the S-wave coherence is very strong, and the converted DAS data compare one-to-one with the nodal seismometers at the start and end of each selected segment. This similarity and waveform coherence clearly manifests itself in the beamforming (Fig. 13f-h): while the original DAS strain rate waveforms

yield ambiguous and complicated beamforming results, the converted DAS and nodal S-wave beamforming results are practically identical, suggesting a well-resolved back-azimuth close to the true back-azimuth of the seismic source and an apparent velocity of $3 \text{ km s}^{-1}$, consistent with previous results (Fig. 5). Hence, we can conclude from this exercise that converting the DAS strain rate to particle velocity alleviates all of the issues associated with the measurement principle pointed out in the previous sections.

## 4   Discussion

### 4.1   Importance of site heterogeneities and seismic wave scattering

The purpose of beamforming is to locate the origin of the incoming signals, which requires that the signals be direct arrivals. As the seismic equivalent of acoustic echoes, scattered waves will follow a trajectory different from the direct arrivals, exhibit a different direction-of-arrival, and so appear to originate from a different source location. Throughout this work, we have

shown that the DAS array recordings at the Brady Hot Springs natural laboratory are strongly affected by scattered waves and local heterogeneities. However, the nodal seismometer array seems much less affected, and displays strong signatures of the Hawthorne earthquake's direct arrivals. To summarise, the likely reasons for this are as follows:

1. For P- and S-waves, DAS strain rate measurements are most sensitive to strains parallel to the direction of the fibre (Martin et al., 2018). Direct arrivals originating from a distant source arrive at the array at a steep inclination, and so

their projection onto the direction of a horizontal fibre is comparatively small (for P-waves, the DAS sensitivity exhibits a $\cos^2$ decay with inclination). By contrast, free-surface topography and shallow subsurface heterogeneities may cause scattered seismic waves to arrive at the array at a shallow inclination, so that these are greatly amplified in the DAS measurements.

2. As pointed out by Daley et al. (2016), the relation between the strain $\varepsilon$ induced by a plane wave and the particle velocity

$\dot{u}$ in the along-fibre direction is $\varepsilon = \pm \frac{1}{c}\dot{u}$, with $c$ being the apparent phase velocity of the medium. This implies that

apparent fast waves (i.e. arriving at steep inclinations) are damped with respect to sub-horizontally-travelling waves such as those generated by shallow scattering.

3. From the numerical simulations and theoretical analysis of Singh et al. (2020), it is immediately clear that gradients in the particle velocity field are highly sensitive to heterogeneities. It has also been observed in real-world DAS experiments (e.g. Jousset et al., 2018; Lindsey et al., 2019) that shallow subsurface features such as faults clearly manifest themselves in the DAS records, further attesting to the sensitivity of DAS strain (rate) measurements to heterogeneities. Since the subsurface beneath the Brady Hot Springs site is strongly heterogeneous (Feigl and the PoroTomo Team, 2018), spatial variations in the phase velocity are expected to exert a strong influence on the DAS measurements. The lack of observable directional sensitivity of the DAS array (our Fig. 2 and Fig. 22 of Wang et al. (2018)) further attest to this. Singh et al. (2020) proposed that a correction term (the "coupling tensor") may be inverted for, so that the amplitudes of the recordings may be adjusted to represent the "true" velocity gradients. Since the true particle velocity field is known (being recorded by the nodal array), this offers an interesting perspective for future analysis of the Brady Hot Springs data set.

4. Lastly, even when considering the directional sensitivity of DAS, the locally recorded strain rates appear to be highly incoherent, while the particle velocity measurements themselves exhibit very strong coherence (compare Fig. 3 with Fig. 8). The superposition of multiple orthogonal components of the particle velocity field may lead to additional destructive interference, even though this was not clearly seen in our analysis in Section 3.3 (see Fig. 8b).

As summarised by Shearer (2015), the effects of phase velocity heterogeneities are most pronounced when the size of the heterogeneity is similar to the seismic wavelength. Owing to a scarcity of large-scale heterogeneities, low frequency signals may be less affected by seismic scattering. Coincidentally, the DAS beamforming results in the lowest frequency band (0.5-1 Hz; Fig. 7) do suggest a source with an azimuth that corresponds with the true back-azimuth, although the spread in the beam pseudo-power is diffuse. This may be a consequence of the signal-to-noise ratio in the lowest frequency range, since the source exhibits low spectral power in this range (Fig. 2). Notwithstanding, DAS has a flat frequency response in strain even at very low frequencies (Lindsey et al., 2020; Paitz et al., 2020), so that further investigations of DAS beamforming at low frequencies are warranted.

## 4.2  Implications for beamforming on sparse and dense DAS arrays

The analysis of the PoroTomo experiment has revealed some limitations of conventional beamforming methods applied to DAS array data, particularly in relation to scattered waves and heterogeneities. However, the Brady Hot Springs geothermal field may be considered a particularly unfavourable setting for DAS seismic beamforming owing to the complexity of the subsurface. Fibre-optical cables deployed on more homogeneous bedrock terranes may not suffer as much from high-amplitude shallow scattering. On the other hand, one of the main promises of DAS is its versatility in deployment conditions, with interesting deployment targets including "heterogeneous" environments such as urban areas (Dou et al., 2017; Fang et al., 2020) and submarine basins (Lindsey et al., 2019; Sladen et al., 2019). For many civil monitoring applications, such as traffic density

monitoring (Liu et al., 2018) and vehicle tracking (Wiesmeyr et al., 2020), some of the issues pointed out in the previous section do not apply, as the signals of interest arrive at the DAS fibre at a shallow (or zero) inclination. However, for the purpose of localising deep or distant sources, the inclination sensitivity of DAS starts to become directly relevant.

A second unfavourable aspect of the PoroTomo dataset is that the DAS array is deployed within a relatively small region (1500 by $500$ m), which limits the resolution of beamforming methods (being proportional to the span of the array). Sparse L-shaped and quasi-linear array configurations provide a much larger array span, at the expense of increased source azimuth ambiguity inherent to linear arrays. As was done in Section 3.4, multiple segments of variable orientations can be combined to resolve this ambiguity. Moreover, the same procedure can be adopted to extract the signals carried by direct arrivals: in the case of the Brady Hot Springs geothermal site, the entire array receives seismic energy from (potentially) multiple nearby scattering sites, obscuring the direct arrivals. By contrast, sparse arrays that extend over long distances may receive seismic energy from different scattering sites along the trace of the cable. By selecting and combining the beam power of several segments following the harmonic averaging method proposed by Rieken and Fuhrmann (2004), sources that are common to all segments are amplified with respect to segment-specific sources (i.e. scatterers), provided that the direct arrivals exhibit a sufficiently large footprint in all segments. Large sparse arrays may therefore be more suitable for seismic beamforming than compact dense arrays.

One key assumption that underlies beamforming is that the signal is carried by a plane wave. This assumption implicitly requires that the source be distant compared to the extent of the array. Moreover, the phase velocity is assumed to be uniform across the array. Both these assumptions are embedded in the definition of the steering vectors through the time delay $\tau$ (see Eq. (1)). Since DAS on fibre-optic cables of several tens of kilometres in length has been demonstrated to be feasible (Lindsey et al., 2019; Sladen et al., 2019), these assumptions may break down for local and regional seismic sources. Moreover, for e.g. earthquake early warning purposes, fault zones may be instrumented with fibre-optic cables running along-strike for many tens of kilometres, so that the finite extent of the rupture and rupture complexity will prevent the application of beamforming methods. Fortunately, the plane wave and uniform velocity assumptions can be relaxed by directly computing the travel time between a candidate source location and a location along the cable, turning the beamforming problem into a back-projection exercise (Kiser and Ishii, 2017; Zhu and Stensrud, 2019). Alternatively, the azimuth estimates derived from beamforming on individual segments can be combined to produce a triangulated source location (e.g. Hutchison and Ghosh, 2017; Stipčević et al., 2017). In both cases, the source localisation results will greatly benefit from the large lateral extent of the DAS arrays.

## 4.3 Hybrid array design

As shown in Section 3.5, converting the DAS strain rates to particle velocities through integration dramatically increases the coherence across the array, and closes the gap between DAS and seismometer arrays in terms of beamforming performance. While at least one seismometer needs to be co-located with the DAS array, this approach motivates the design of "hybrid" arrays that comprise both DAS and seismometers. We here propose three such hybrid arrays (Fig. 14) that optimally employ the available seismic stations while reducing the length of DAS cable used in the array:

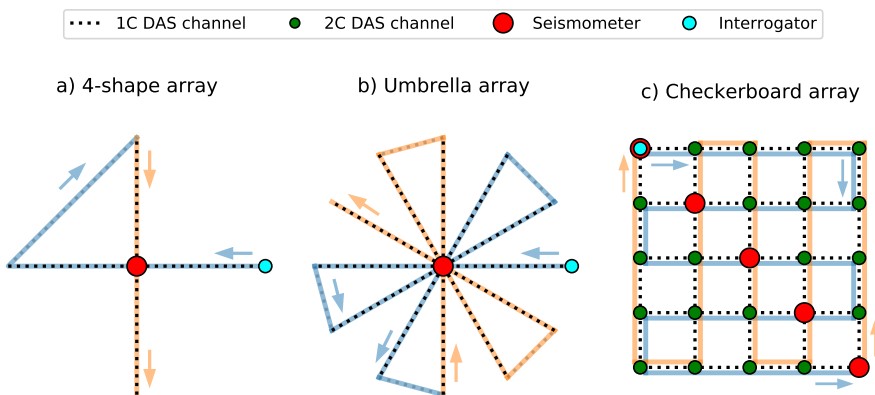

**Figure 14.** Three proposed array configurations that efficiently combine seismometers with DAS. a) A 4-shaped array turns a single seismometer into a seismic array by integrating the DAS strain rates along the east-west and north-south segments; b) An umbrella-shaped array acknowledges the DAS directional sensitivity and features multiple directions along which the strain rates are integrated; c) By deploying the DAS cable in a grid with seismometers along the diagonal, two-component measurements of the velocity field can be obtained at the intersection of perpendicular DAS segments. For reference, the direction of the DAS cable deployment is indicated, with the first half of the cable indicated in light blue and the second half of the cable in orange, starting at the interrogator.

1. A minimal array configuration that records the two independent horizontal components of the particle velocity field features a single seismometer at the centre of a number 4-shaped array (Fig. 14a). The strain rates recorded by DAS are integrated in the directions away from the central seismometer to give one-component measurements of the wavefield in the east-west or north-south direction. The diagonal segment that connects the extremities of the array is not directly connected to a reference station, and so it cannot be integrated to yield particle velocities in this direction. Nonetheless, this connecting segment can be identified and ignored in the analysis, focusing solely on the orthogonal segments connected to the central seismometer. The total length of cable used in this configuration is $\left(4 + \sqrt{2}\right) L$, where $L$ is the segment length from the centre to one of the corners of the array.

2. Since DAS measurements have a distinct directional sensitivity to various types of ground motions (Martin et al., 2018), two independent directions of integration may not be sufficient for the analysis of seismic sources of arbitrary back-azimuth. To improve the likelihood of having a DAS segment that is optimally oriented for a given back-azimuth, a single DAS cable can be deployed to form multiple segments at fixed orientation increments (Fig. 14b). This umbrella-shaped array is a generalisation of the 4-shaped array shown in Fig. 14a, and it could feature an arbitrary number of radial segments. The example shown in Fig. 14b features six independent segment orientations. The total DAS cable length for an umbrella array with $n$ independent segments (i.e. $n = 2$ in Fig. 14a and $n = 6$ in Fig. 14b) is approximately $2L\left(n + \pi\left[n - 1\right]/2n\right)$.

3. In the array configurations using only a single seismometer, integration along each DAS segment yields at most one component of the particle velocity field for that given segment. When multiple seismometers are available, the DAS cable can be deployed in a grid-like fashion with the seismometers positioned along the diagonal of the grid – see Fig. 14c. At the locations where two DAS segments intersect at right angles, the two independent horizontal components of the particle velocity field can be inferred. While other configurations of seismometer placement are possible (e.g. along the boundaries of the checkerboard array), placing the seismometers along a diagonal yields the optimal number of DAS cable intersections per seismometer (the total number of DAS intersections is given as $n^2 - n$, with $n$ being the number of seismometers). Correspondingly, the total length of DAS cable required for this array design is $2\Delta x \left(n^2 - 1\right)$, $\Delta x$ being the grid spacing.

Owing to attenuation of the scattered light intensity with increasing cable length, standard commercial fibre-optic cables permit a maximum sensing distance of roughly 50 km. Taking this as the maximum length $L$ of the cable available for the array, and taking a grid spacing of 200 m, a checkerboard grid can be constructed with $n = \text{floor}\left(\sqrt{\frac{L}{2\Delta x} + 1}\right) = 11$ seismometers. Consequently, this array will have a total span of $(n-1)\Delta x = 2\,\text{km}$, with $n^2 = 121$ two-component measurements every 200 m. For a grid spacing of $\Delta x = 20\,\text{m}$ (ignoring the limitation of the gauge length for the moment), the corresponding number of two-component measurements and total array span are $35^2 = 1225$ and 700 m, respectively. For comparison, this array span and density are similar to the San Jacinto Fault array that was deployed by Roux et al. (2016), which featured 1108 single-component geophones distributed over a $650 \times 700\,\text{m}^2$ area. But instead of requiring 1108 individual stations, the hybrid array only requires 35 seismic stations, which dramatically reduces the hardware costs for such an operation. From this calculation it is apparent that there is a trade-off between the extent of the array (which roughly scales as $\sqrt{\Delta x}$) and the spatial density of the measurements (which roughly scales as $1/\Delta x$). The optimal spacing of the measurements must therefore be chosen appropriately for the application.

The complexity of deployment of a fibre-optic array is higher than for a temporary nodal array due to the need for trenching. However, since permanent seismometers are typically installed in shallow holes in the ground, the complexity and challenges of deploying a hybrid DAS-seismometer array is likely similar to those of a permanent seismometer array. This renders hybrid array designs with strategically deployed seismometers and DAS cable segments a cost-efficient alternative to permanent seismometer arrays, facilitating many applications including microseismicity monitoring, ambient noise tomography, and seismic source characterisation. One particular scenario that we wish to highlight here is that of rapid aftershock monitoring: the deployment of seismic stations following a large earthquake is time consuming and logistically challenging, and most often the temporary deployments are not completed in time to capture the earliest stage of the aftershock sequence. In contrast, fibre-optic cables can be deployed permanently in the 4-shape or umbrella array configurations around permanent seismic stations in earthquake-prone regions, and, as soon as an earthquake of interest occurs, an interrogator unit may be readily connected to record the very earliest stages of the aftershock sequence.

## 5 Conclusions

This study considered the potential of fibre-optic Distributed Acoustic Sensing (DAS) arrays for the purpose of seismic beamforming. This was done by performing beamforming on the ground motions generated by the March 2016 $M_L$ 4.3 Hawthorne earthquake, as recorded by a DAS array co-located with a dense nodal seismometer array at the Brady Hot Springs geothermal field. Comparing the waveforms recorded by DAS with those recorded by the nodal seismometers, we find that the strong waveform coherence of the nodal array is absent in the DAS array. Since the quality of the beamforming results depends strongly on waveform coherence, the DAS array is unable to produce a robust source azimuth and apparent velocity, whereas the nodal array produces an extremely well-resolved source location that is consistent with the true back-azimuth of the earthquake epicentre. Instead, beamforming on the DAS array reveals source locations that likely correspond with shallow seismic scattering sites. We attribute the lack of DAS waveform coherence to the DAS measurement principle, which inherently leads to diminished sensitivity of DAS recordings to the direct arrivals, supposedly arriving at the array at a steep inclination, and amplifies scattered waves arriving at shallow inclinations. Moreover, we demonstrate that the spatial gradients of the particle velocity field (i.e. strain rate) exhibit far lower coherence than do the particle velocity waveforms, which additionally impedes beamforming. Compared to other DAS arrays, this may be aggravated by the strong phase velocity heterogeneities present at the Brady Hot Springs geothermal field. Fortunately, all of the above issues can be alleviated by converting the DAS strain rate measurements into particle velocities, taking the recordings of a nodal seismometer as a reference point and incrementally integrating the strain rates along a linear DAS segment. This approach warrants the design of "hybrid" arrays comprising both DAS and nodal seismometer arrays.

*Code and data availability.* Python scripts that reproduces the results and figures in this manuscript are available at https://doi.org/10.6084/m9.figshare.12899288.

*Author contributions.* MvdE conceptualised the study and performed the analyses. JPA supervised MvdE and contributed to methodology and interpretation. Both authors discussed and prepared the contents of the manuscript.

*Competing interests.* The authors declare no competing interests

*Acknowledgements.* This work was supported by the French government through the UCA[JEDI] Investments in the Future project managed by the National Research Agency (ANR) with the reference number ANR-15-IDEX-01. The authors thank Lingsen Meng for sharing his MUSIC back-projection codes, from which the beamforming codes used in this study were derived. The nodal seismometer data were processed using ObsPy (Beyreuther et al., 2010), and generic data manipulations were performed with NumPy (Harris et al., 2020). Data

visualisation was done using Matplotlib (Hunter, 2007). The authors are grateful for the constructive comments of the handling editor P. Jousset, two anonymous reviewers, and E. Martin.

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
