# Peer review of "Evaluating Seismic Beamforming Capabilities of Distributed Acoustic Sensing Arrays"

_Solid Earth, 2020_

## Short Comment (SC1) · 25 Sep 2020

This point of spatial directional derivatives leading to higher influence of lateral heterogeneities makes sense. You went in one direction to show this (node -> DAS equivalent). If you convert to single component velocity equivalent (described in Wang et al 2018) before applying MUSIC beamforming, is this enough to noticeably reduce the influence of those heterogeneities?

Looking forward (beyond scope of this paper), you may want to get in touch with Siyuan Yuan, a student at Stanford. He did a technical report a few years ago on a method to improve simple beamforming with adjoint wave operators on DAS data, and he ran into

some similar-looking artifacts in the process of that work. Ultimately, he ended up with separate corrections for P/SV and SH waves depending on the direction of fiber, and a few other optimization techniques to improve the results. I'm not sure if that work got turned into a paper, but it could be a good starting point for finding some ways to adjust MUSIC to work better.

Thanks for an interesting paper! Eileen Martin (Virginia Tech)

---

## Referee Comment (RC1) · Anonymous Referee #1 · 25 Oct 2020

This paper focuses on discussing the applicability of beamforming to DAS by comparing co-located seismometers in the Brady geothermal field. Authors showed different results from DAS array and seismometer array from one earthquake. Authors discussed the potential problems from the DAS strain rate measurements. Finally, the local heterogeneity could be the major source of this inaccuracy.

1. Line 53, please add the focal depth information. And why this earthquake?

2. Section 2.2, Any further explanation why chooses MUSIC. It's confusing that you describe a lot about classical beamforming approaches and then say you used MUSIC, with very less details to follow. Could you elaborate the MUSIC method here and why

this is the choice here over classical beamforming.

3. Fig 3 shows P wave but why there are two arrivals (17.5s and 22.5s)? What's the phase of second arrival if it is not S wave? And please add a line to help show the arrival time difference across the array.

4. Followed by comment 3, From figure 3 the waveforms from Z-component are strong. Please add explanation why the beamforming results from Z component in the frequency band 0.5-1 Hz is not good. Is this because the wavelength is close to the scale of array?

5. Please clarify the time window length of data sections used for beamforming. Since you mention many scattered P waves, will different time windows improve the results, shorter window with fewer phases? For example, just choose a few seconds (4s maybe?) of recordings around the P-arrivals that only include the first arrivals.

6. figure 6 is hard to read (where is P and S?) and possibly misleading. According to the geometry of fiber, DAS data is strain rate recordings of many directions. Figure 6 shows horizontal seismic data from all azimuths. The polarity of S-wave could be flipped at each fibre corner. Could you try the image display instead of wiggles?

Line 134, "polarity flips are anticipated", I want to remind that this is true for shear waves, but may not true for other phases. This could be very important to correct for S wave before doing beamforming. This may be the reason of the diffusion of the focus in Figure 7.

I am curious whether a short "L" shape segment can be demonstrated.

Regarding to using long DAS array to locate the source, "the source localisation results will greatly benefit from the large lateral extent of the DAS arrays", I refer you to look at recent publication by Zhu and Stensrud ,2019 to backpropagate full waveform DAS data to locate the source.

---

## Referee Comment (RC2) · Anonymous Referee #2 · 31 Oct 2020

General overview:

This work aims to explore the performance of DAS array for the purpose of beamforming, in comparison with a co-located nodal seismometer array. The authors used ground motions generated by a ML 4.3 earthquake that occurred on 21 March 2016 and located 150 km of the geothermal field of Brady Hot Springs. The Poroelasctic Tomography (PoroTomo) project conducted an experiment on this geothermal site in March 2016, involving an array of 238 3C seismometers deployed over an area spanning 1500 by 500 m, as well as several fibre-optic cables for DAS sensing. After a presentation of the PoroTomo experiment and an introduction of the MUSIC beamforming method

used in this work, spectral characteristics of the signal recorded by nodal sensors and DAS are presented showing that DAS has a flat frequency response even at very low frequencies and is characterized by the lack of directional sensitivity.

Beamforming analysis applied to nodal arrays with P- and S-waves shows a well-resolved source with a back-azimuth close to the true back-azimuth. This result is consistent with a very strong coherence of waveforms across the entire nodal array. In strong contrast with the nodal array, P- and S-waves recorded by the DAS show a low degree of coherence, which means that DAS array is unable to produce a robust source back-azimuth and apparent velocity. By simulating DAS recordings from nodal array, the authors show that the lack of coherence and beam resolution seen in the DAS data cannot be attributed to DAS-specific technicalities like coupling of the DAS cable with the ground. This kind of analysis is very useful and informative for the DAS community. However, clarifications should be provided. To compare the strain rate obtained with the nodes to that measured with the DAS, shouldn't we use a distance between the nodes equivalent to the gauge length? This is not clear to me and it is important to clarify this point in the article.

The authors then selected individual small, linear segments of the DAS array, which exhibit high waveform coherence. Combining the beamforming results of these segments, they obtained results in better agreement with the true arrival of the seismic source. Nevertheless, this work and theses analysis show that beamforming methods that are traditionally applied to particle velocities don't give so good results when applied to strain rates. DAS array reveals source locations that likely correspond with shallow scattering sites. In conclusion, authors state that the waveform low coherence is inherent to DAS measurement principle, which leads to lower sensitivity of DAS recordings to waves arriving at the array with a steep inclination, and amplifies scattered waves arriving at shallow inclinations.

This article is very well written and presented, the data analysis is clearly outlined, even if some clarifications have to be provided. This is probably one of the first analysis comparing results of beamforming analysis obtained with strain rate recorded with a DAS and particle velocities recorded with a co-located nodal array. The contribution of this work is significant because it constitutes an advance in the better understanding of DAS recordings. It also constitutes an important contribution from the point of view of beamforming analysis of strain rate data. This work shows the limits of the application of beamforming methodology to strain rate data as it is usually applied to particle velocities recorded with inertial seismometers and this constitutes a very useful information for the seismological community. From my point of view, this work deserves to be published. I have some minor remarks in the following.

Minor comments:

3.2 Beamforming results of the nodal and DAS arrays

- Line 117: Indicate is beamforming was performed by using all the sensors. An array response function could be useful to see the influence of the geometry of the array. As the number of seismometers is huge, I am wondering if you tried to analyze data recorded by sub-arrays.

- Lines 125-126: "(consistent with the ratio of vertical to horizontal amplitudes of the nodal P-waveforms)". This sentence is not clear to me. Amplitudes of vertical and horizontal components are not commented in the text.

- Line 134: I don't understand what "polarity flips" mean in this context. Can you clarify ?

- Line 138: It is not clear if all the fibre-optic cable was used for the beamforming analysis. The gauge length should be indicated, as well as the distance between each strain rate measurement and the total number of individual SR measurements used in the beamforming analysis.

3.3 Simulating DAS recordings from the nodal array

-Line 152: one component of strain

- Line 166: The angle teta is considered relatively to North ?

- Line 155-171: The simulated strain rate is calculated with node pairs separated by a distance less than 80m. Is this distance equal or bigger than the gauge length used during the experiment (the GL used during the experiment is not indicated in the manuscript)? Why not showing in a figure simulated strain rate signal and measured strain rate on the same segment?

3.4 Selective beamforming of the DAS array

The beamforming analysis on the P-waveforms recorded by the DAS array should be clarified. Are you using the same analysis parameters as for the beamforming of the nodal array? In term of frequency range, time window length? It would be useful to indicate how long are the segments of fibre used for beamforming analysis and how many strain rate measurement points they include.

- Line 233-234: Indicate in parentheses the segments corresponding to a direction-of-arrival between the east and the south and a direction-of-arrival from the north. The text will be easier to read if you indicate the number of the segments.

- Line 233-234 and figure 12: What is discriminating more between the 2 groups of segments, the back-azimuth or the apparent velocity ?

- Line 240: "implicitly"

- Line 245: As mentioned before, indicate the number of the segments.

- Line 245: When excluding segments 4 and 6, the geometry of the DAS array used for beamforming is relatively linear. This geometry probably explain the elongating shape of the PDF in figure 12d. As I mentioned before for Line 117, it would be useful to show the array response function.

Have you tried to estimate the slowness vector with sub-arrays? For example, segments 1, 2 and 3 constitute an array with a less linear geometry.

Figures:

Figure 8: The indication of the direction of propagation of the wavefield would help to better understand the selection of the segments in red.

Figures of beamforming results: Apparent velocity values are difficult to read, either because they are represented in red, or because character size is too small.

---

## Author Comment (AC1) · 14 Dec 2020

**[R1.1]** Line 53, please add the focal depth information. And why this earthquake?

The focal depth of 9.9km has been added. In the original manuscript, we mentioned in lines 59-61 that "*we retrieve and interpret the same data set as was analysed by Wang et al. (2018), and so we build upon the conclusions drawn from this previous study*", which motivates the choice for this earthquake.

**[R1.2]** Section 2.2, Any further explanation why chooses MUSIC. It's confusing that you describe a lot about classical beamforming approaches and then say you used MUSIC, with very less details to follow. Could you elaborate the MUSIC method here and why this is the choice here over classical beamforming.

In the original manuscript, we motivate the choice for MUSIC as: "*[MUSIC] is an extension of classical beamforming approaches that acknowledges sparsity in the number of signals arriving at the array, resulting in higher-resolution estimates of the back-azimuth and slowness of the seismic waves*" (lines 82-84). To clarify the distinction from delay-and-sum beamforming, we have now added to lines 93-95:

"*The procedure of estimating $C^2$ is as described above, and so the sole difference between MUSIC and traditional beamforming lies in the projection of the steering vectors onto the noise space (and taking the reciprocal), rather than projecting onto the full space of $C^2$*"

For a derivation of the MUSIC method, we refer to the original work of Schmidt (1986).

**[R1.3]** Fig 3 shows P wave but why there are two arrivals (17.5s and 22.5s)? What's the phase of second arrival if it is not S wave? And please add a line to help show the arrival time difference across the array.

Given the similarity with the first arriving phase (strong N and Z component), it is likely that the second phase at around 21s is a second P-arrival. One could speculate that this is a scattered wave originating from a scattering site close to the seismic source or a depth phase (a reflection from the Earth's surface), but we have no direct evidence for that.

A line to show the move-out across the array has been added to each panel.

**[R1.4]** Followed by comment 3, From figure 3 the waveforms from Z-component are strong. Please add explanation why the beamforming results from Z component in the frequency band 0.5-1 Hz is not good. Is this because the wavelength is close to the scale of array?

The wavelength at 1Hz is of the order of 5 km, while the span of the array is only 1.5 km, so it would seem plausible that the limit of the array resolution is observed in Fig. 4. However, the source for a 1-2 Hz frequency band is very well resolved, while the wavelength (> 2.5 km) still exceeds the span of the array. We therefore speculate in lines 128-130 that:

"*Only in the 0.5-1.0 Hz frequency band does the beamforming of this component lead to a relatively poorly resolved location, which may be due to the influence of the corner frequency (typically around 1-3 Hz for an ML 4.3 event; see e.g. Scholz (2019)).*"

**[R1.5]** Please clarify the time window length of data sections used for beamforming. Since you mention many scattered P waves, will different time windows improve the results, shorter window with fewer phases? For example, just choose a few seconds (4s maybe?) of recordings around the P-arrivals that only include the first arrivals.

We now mention in lines 118-119 that:

"*We take a time window from 2s before to 8s after the first arrival of each respective phase (i.e. 10s in total).*"

We have experimented extensively with various time windows, and found that the DAS beamforming results did not improve when choosing a window tightly centred around the first arrival. This is also seen in the newly added Figure 13.

**[R1.6]** figure 6 is hard to read (where is P and S?) and possibly misleading. According to the geometry of fiber, DAS data is strain rate recordings of many directions. Figure 6 shows horizontal seismic data from all azimuths. The polarity of S-wave could be flipped at each fibre corner. Could you try the image display instead of wiggles?

With Fig. 6 we aimed to offer a similar representation as for the nodal array (Fig. 3), i.e. wiggles ordered by distance from the seismic source. These wiggles are separately plotted for the P- and S-waves, so there is no question as to which phases are represented. We fully agree that DAS records a mixture of horizontal components, and showing the overall incoherence of the signals is precisely the point of this figure. When the waveforms are plotted as an image display, this incoherence is not so clearly observed.

**[R1.7]** Line 134, "polarity flips are anticipated", I want to remind that this is true for shear waves, but may not true for other phases. This could be very important to correct for S wave before doing beamforming. This may be the reason of the diffusion of the focus in Figure 7.

We have now specified in this line that the polarity flips are anticipated only for S-waves. The diffuse spread is equally large when beamforming the P- and S-phases, so we do not attribute the spread at the lower frequency bands to changes in S-wave polarity.

**[R1.8]** I am curious whether a short "L" shape segment can be demonstrated.

Owing to the layout of the cable at Brady Hot Springs, there are few perpendicular segments, all of which are relatively small in extent (of the order of 100m), and thus exhibit poor beamforming resolution. Moreover, these small segments or subarrays likely receive energy from a single scattering location, dominating the beamforming results. When mentioning L-shaped arrays in Section 4.2, we had in mind that these "sparse" arrays can be of much larger extent (of the order of tens of kilometres), and so the local scattering can be better distinguished from the direct arrivals that are common to the entire array.

**[R1.9]** Regarding to using long DAS array to locate the source, "the source localisation results will greatly benefit from the large lateral extent of the DAS arrays", I refer you to look at recent publication by Zhu and Stensrud, 2019 to backpropagate full waveform DAS data to locate the source.

We thank the reviewer for bringing this interesting study to our attention for future work. We now also include this reference in Section 4.2, line 399.

---

## Author Comment (AC2) · 14 Dec 2020

**[R2.1]** To compare the strain rate obtained with the nodes to that measured with the DAS, shouldn't we use a distance between the nodes equivalent to the gauge length? This is not clear to me and it is important to clarify this point in the article.

Following Wang et al. (2018), the average DAS-recorded strain rate between two nodes, separated by a distance 2L, is exactly given by the difference between the waveforms of the two nodes divided by 2L. This relation is exact for arbitrary L. We now also mention in lines 175-176 that this relation holds for separation distances larger than the gauge length:

*"A similar relation holds when the distance between the nodes is a multiple of the DAS gauge length (see Wang et al., 2018, their Eq. 5)"*

Minor comments:

*3.2 Beamforming results of the nodal and DAS arrays*

**[R2.2]** Line 117: Indicate is beamforming was performed by using all the sensors. An array response function could be useful to see the influence of the geometry of the array. As the number of seismometers is huge, I am wondering if you tried to analyze data recorded by sub-arrays.

We have added a mention that all the sensors in the array are used in the beamforming. Since the seismic source radius of an M4.3 earthquake is small compared to the Rayleigh diffraction limit, it can be well approximated by a point source. Moreover, the source radius is of the same order as the Fresnel zone size, which again leads to the conclusion that the source can be approximated by a point source. Hence the beampatterns obtained in this study are good approximations of the nodal array response.

We now analyse a subset of the DAS array segments in the newly added Section 3.5 (see also Fig. 13a in this section).

**[R2.3]** Lines 125-126: "(consistent with the ratio of vertical to horizontal amplitudes of the nodal P-waveforms)". This sentence is not clear to me. Amplitudes of vertical and horizontal components are not commented in the text.

As a quick check, we measured the ratio of the vertical to horizontal component of the P-waveform recorded by the nodal seismometers, which is indicative of the inclination of the P-wavefield. Since we performed this check only as a back-of-the-envelope verification of the inclination estimated from the beamforming results, we chose not to discuss this aspect in more detail.

**[R2.4]** Line 134: I don't understand what "polarity flips" mean in this context. Can you clarify?

We clarified the polarity flips as:

*"Thirdly, depending on the orientation of the fibre, S-wave polarity flips are anticipated (Fang et al., 2020). These polarity flips are due to the projection of the particle motion onto the fibre, leading to contraction in some segments and extension in others (Lindsey et al., 2017)."*

**[R2.5]** Line 138: It is not clear if all the fibre-optic cable was used for the beamforming analysis. The gauge length should be indicated, as well as the distance between each strain rate measurement and the total number of individual SR measurements used in the beamforming analysis.

We now mention in lines 52-54 that: "*The gauge length was taken to be 10m, which was supersampled to give a channel spacing of 1m (i.e. one strain rate measurement was made every 1m)*".

In lines 150 we now state that we perform the beamforming on "*the entire DAS array recordings (8621 channels in total)*".

*3.3 Simulating DAS recordings from the nodal array*

**[R2.6]** Line 152: one component of strain

Corrected

**[R2.7]** Line 166: The angle theta is considered relatively to North?

We now mention in this line that the angle is measured relative to east.

**[R2.8]** Line 155-171: The simulated strain rate is calculated with node pairs separated by a distance less than 80m. Is this distance equal or bigger than the gauge length used during the experiment (the GL used during the experiment is not indicated in the manuscript)? Why not showing in a figure simulated strain rate signal and measured strain rate on the same segment?

We now indicate in the manuscript that the gauge length is 10m. For (several examples of) a comparison between measured strain rates and those derived from a pair of nodal seismometers, we refer in the manuscript to Wang et al. (2018).

*3.4 Selective beamforming of the DAS array*

**[R2.9]** The beamforming analysis on the P-waveforms recorded by the DAS array should be clarified. Are you using the same analysis parameters as for the beamforming of the nodal array? In term of frequency range, time window length? It would be useful to indicate how long are the segments of fibre used for beamforming analysis and how many strain rate measurement points they include.

We now state in lines 150-151:

"*When we nonetheless continue to perform beamforming on the entire DAS array recordings (8621 channels in total), we obtain highly variable results (Fig. 7) for the same window length and frequency range as was used for the nodal array.*"

The beamforming is performed on the entire DAS array at once, and not segment-by-segment. The length of individual segments is therefore not directly relevant.

**[R2.10]** Line 233-234: Indicate in parentheses the segments corresponding to a direction-of-arrival between the east and the south and a direction-of-arrival from the north. The text will be easier to read if you indicate the number of the segments.

We now explicitly indicate the segment numbers with a direction east/south and those with a direction north.

**[R2.11]** Line 233-234 and figure 12: What is discriminating more between the 2 groups of segments, the back-azimuth or the apparent velocity?

In line 245 of the original manuscript (line 261 in the revised manuscript), we mention that "*When we exclude the two segments with an apparent source north of the array*", i.e. the discriminating feature is the back-azimuth of the source inferred for each segment.

**[R2.12]** Line 240: "implicitly"

Corrected.

**[R2.13]** Line 245: As mentioned before, indicate the number of the segments.

We added that "*four segments remain*" in this line.

**[R2.14]** Line 245: When excluding segments 4 and 6, the geometry of the DAS array used for beamforming is relatively linear. This geometry probably explain the elongating shape of the PDF in figure 12d. As I mentioned before for Line 117, it would be useful to show the array response function.

The reviewer is correct that when combining multiple linear segments with a similar orientation, the intersection of their beams is less well resolved, resulting in a broader spread of beampower. Fortunately, segment 5 exhibits a slightly different orientation from segments 1-3, which alleviates this somewhat.

Considering the extent of the DAS array with respect to the extent of the seismic source and its distance from the array, the distribution of beampower in Fig. 12 is representative for the array response (see also our reply to comment R2.2).

**[R2.15]** Have you tried to estimate the slowness vector with sub-arrays? For example, segments 1, 2 and 3 constitute an array with a less linear geometry.

In this particular section, we aimed to provide an automated segment selection criterion based on the internal coherence of the segments (rather than manually and arbitrarily selecting segments). We have not explored automated selection criteria for combining multiple segments into (non-linear) subarrays. When we manually pick and combine segments 1, 2, and 3 into a subarray and perform the beamforming, we get a similar result as what was obtained in Fig. 12d of the manuscript, though with a much higher slowness (just over 1 s/km; see the figure included below,

which shows the slowness space for the 0.5-1 Hz frequency band). This result does not contribute much to the overall conclusions of this section, and more systematic methods of creating subarrays need to be explored.

[Figure]

*Figures:*

**[R2.16]** Figure 8: The indication of the direction of propagation of the wavefield would help to better understand the selection of the segments in red.

The propagation direction has been added.

**[R2.17]** Figures of beamforming results: Apparent velocity values are difficult to read, either because they are represented in red, or because character size is too small.

We agree with the reviewer that the radial axis labels are hard to read on paper and on small screens. We have experimented with different colours and sizes, and we concluded that the current representation is the clearest. A larger font size or bold face will lead to crammed/overlapping numbers, and other colours have insufficient contrast to the chosen colourmap (or to the white background of the image). Fortunately, the vectorised PDF images can be enlarged on-screen without loss of fidelity.

---

## Author Comment (AC3) · 14 Dec 2020

We thank dr. Martin for her suggestions for the manuscript. Converting the DAS strain rates to a horizontal velocity component is indeed an intuitive step to make, and in fact we explored this idea when this manuscript was first submitted. We have now completed the analysis, and we have dedicated a new section (3.5) to describe the procedure and results of integrating the DAS strain rates. In summary, performing this analysis leads to DAS beamforming results that are nearly identical to those of the nodal array, and are thus better (less affected by heterogeneities) than beamforming the raw DAS strain rate data.

---

## Author Response (AR2)

We received the reviewer report directly from the Topical Editor. Below we respond to the reviewer's queries in a point-wise fashion:

**[R1.1]** *When converting strain rate to particle velocity, I didn't see any velocity scaling. So the waveform in Fig 13e is strain? Please clarify this.*

No "velocity scaling" is needed in our method; the procedure is encapsulated in Eq. (7). The waveforms in Fig 13e are velocity, not strain. In lines 281-282 we added that: "*[...] no assumptions are made regarding the apparent propagation speed of the signals to convert strain rate into particle motion (as is typically done in other methods; e.g. Lior et al., 2021; Zhu et al., 2021, this issue)*". This convenient result follows from Eq. (7).

**[R1.2]** *In section 3.2, there are several reasons to explain the DAS waveform incoherence. In the conclusion part, the author mainly ascribes the failure of DAS beamforming to the incoherence of DAS traces caused by local scattering (line 463). Maybe I was missing why the scattering is the main reason over others, like axial sensitivity of optic fiber? Fig 10 shows the pretty good coherent waveforms in each linear segment, which seems that scatterings are not strong.*

In Section 3.4 of the original manuscript, we investigate these segments with locally strong waveform coherence, and we conclude that these coherent signals likely originate from scattering sources. So even though locally coherent segments can be found, this coherence does not persist at the scale of the array.

---

## Author Response (AR3)

[1] *Thanks for your patience. I reread in detail the final version of your manuscript and I noticed that the section 3.5 is entirely new, and therefore was not reviewed by anyone.*

*I am sorry that there things that I do not understand in this section. Note I asked advise to one of the co-editor for double checking (G. Currenti). It is not clear at all how you can go from a linear strain rate record to a "tensorial form" of the strain rate. A tensor (assuming 2D) takes into account strain (rate) components in 2 directions, with off diagonal components. However, as you mention in your manuscript line 292, DAS measure only one component in the direction of the fibre. It is indeed only the projection of the strain tensor onto the DAS inline direction. It unfortunately not possible to go back to the strain (rate) tensor (3 terms in 2D isotropic elastic) from the DAS measurements (1 direction including all components of the tensor projected). Therefore the equation line 299 is very confusing, and I would strongly advise you to reconsider, both the phrasing and the equation that you can derive from a single direction measurement. Moreover, I do not understand what you mean by "normal vector".*

We do not claim that we can retrieve the strain tensor from DAS measurements. We apologise for the confusing sentence and notation. The subscripts $ij$ were intended to refer to spatial locations $x_i$ and $x_j$, and not to the directional components of the strain tensor. We have clarified this in the revised manuscript by omitting the subscripts on the left-hand side and by explicitly stating that this is the longitudinal (along-cable) strain rate. We now refer to the "normal" vector n as the "unit" vector pointing from $x_j$ to $x_i$. We hope that with these modifications this sentence has become sufficiently clear. It now reads:

"The longitudinal (along-cable) strain rate recorded by DAS is $\dot{\varepsilon} = \frac{1}{L}\left(\dot{\mathbf{u}}(x_j) - \dot{\mathbf{u}}(x_i)\right) \cdot \mathbf{n}_{ij}$, where $\mathbf{n}_{ij}$ is the normal unit vector pointing from $x_j$ to $x_i$."

[2] *In general, could you state that integration of strain rate to strain would give better beam-forming results than strain rate? This is actually the same as performing beam-forming on velocities that on acceleration.*

Since we only perform narrow-band beamforming, using strain or strain rate makes no difference on the beamforming results. Compared to narrow-band beamforming, performing broadband beamforming on strain instead of strain rate is equivalent to weighting by 1/frequency, so that higher frequencies contribute less to the overall results. In some situations this might be desirable (for instance, to de-emphasise incoherent high-frequencies), but this should be evaluated on a case-by-case basis.

Minors other points:

[3] *Line 119. There is a blank missing before "To visualize…"*

Corrected.

[4] *Line 285. You indicate that the gauge length is L here, but actually at line 174 the gauge length is defined as being 2L. This is inconsistent and need to be arranged.*

We now use L to indicate the gauge length.

**[5]** *Line 336. minor point. You did not really demonstrated that local heterogeneity are modifying the strain rate. You used previous results from Wang et al., 2018 for instance as assumption. In addition, "Scattering" appears in the introduction and in the discussion, but never in the core of the text, so I guess you cannot claim you demonstrated that scattering is the reason. You rather processed data to remove the effect of scattering shown by others. The same applies for heterogeneity, as you point out actually at line 467.*

In the results section of the original manuscript, we identify scattered waves predominantly by their low apparent velocity, which suggest a shallow angle of incidence rather than a steeply-inclined angle of incidence expected for the direct arrivals of a deep source (an earthquake). Since these scattering sites may lie outside of the span of the array, they cannot be directly imaged. However, indirect evidence for scattering is presented in various locations in the results section. Since the scattered waves are expected to exhibit a low propagation speed, their presence in DAS recordings is amplified. Combined with the numerical observations and analysis of Singh et al. (2020), we believe to have presented a plausible case for the role of scattering and heterogeneities in the medium surrounding the DAS array.

**[6]** *Line 365. DAS response is flat in strain, but not is strain rate.*

This has been added to line 365.

**[7]** *Line 440-445. Interesting approach, however, standard commercial cables do not necessarily follow the ideal theoretical frame. So you could mention how to cope with this issue, as deploying cables is expensive, and less easy than seismometers (need to dig a trench etc).*

For this section we only had dedicated cable deployments in mind, and not tapping into existing commercial cables. We have added the following note on the deployment in lines 449-454 of the revised manuscript:

"*The complexity of deployment of a fibre-optic array is higher than for a temporary nodal array due to the need for trenching. However, since permanent seismometers are typically installed in shallow holes in the ground, the complexity and challenges of deploying a hybrid DAS-seismometer array is likely similar to those of a permanent seismometer array. This renders hybrid array designs with strategically deployed seismometers and DAS cable segments a cost-efficient alternative to permanent seismometer arrays, facilitating many applications including […].*"

We also expanded on the application of aftershock monitoring as follows (lines 455-459):

"*[…] and most often the temporary deployments are not completed in time to capture the earliest stage of the aftershock sequence. In contrast, fibre-optic cables can be deployed permanently in the 4-shape or umbrella array configurations around permanent seismic stations in earthquake-prone regions, and, as soon as an earthquake of interest occurs, an interrogator unit may be readily connected to record the very earliest stages of the aftershock sequence.*"

**[8]** *Line 553. The title of this paper is written twice.*

Corrected.

Non-public comments to the Author:

**[9]** *For completeness, I would then suggest to add a reference to Jousset et al., 2018. Below some suggestions where to cite it. Not all suggestions are necessary!*

We have added the reference to Jousset et al.

To briefly comment on the approach of Jousset et al. (2018) as compared to ours, we would like to point out that the space-time integration of DAS strain rates as performed by Jousset et al. yields displacements relative to the starting point of integration (p.8 of Jousset et al.: "*If we integrate the local strain with respect to space, the **relative** displacement can be calculated at all points along the profile*"). In our proposed method we use a reference seismometer at one end point to obtain absolute ground motions, not relative ones.

**[10]** *Just an additional note: as data is available (open access) for the Iceland array with both seismometers and fibre optic cables data, it would be nice to test your approach as and see if it works or not for the frequencies given for a local earthquake, and the differences and limitation of your approach. However, I am fully aware that is too long, and could make additional paper.*

We appreciate the suggestion. We are collaborating with Fabian Walter to perform an in-depth analysis of the limitations of the proposed technique, using data from geophones and a long stretch of fibre deployed on a glacier. We will also consider the data from Iceland to investigate the performance of the method in a different setting.

---

## Author Response (AR4)

Dear editor,

We appreciate the critical evaluation of Eq. (8) and subsequent paragraphs. Based on your comments, we realised that the procedure for the sign correction as described in the text does not accurately reflect the procedure adopted to generate the results of Fig. 13. After careful deliberation, we have reformulated lines 296-306 of the revised manuscript in a way that we believe now accurately describes the procedure.